# Dynamic climate-driven controls on the deposition of the Kimmeridge Clay Formation in the Cleveland Basin, Yorkshire, UK

Elizabeth Atar[1], Christian März[2], Andrew Aplin[1], Olaf Dellwig[3], Liam Herringshaw[4], Violaine Lamoureux-Var[5], Melanie J. Leng[6], Bernhard Schnetger[7], Thomas Wagner[8].

[1] Department of Earth Sciences, Durham University, South Road, Durham, DH1 3LE, UK
[2] School of Earth and Environment, University of Leeds, Leeds, LS2 9JT, UK
[3] Leibniz-Institute for Baltic Sea Research, Marine Geology, Seestrasse 15, 18119 Rostock, Germany
[4] School of Environmental Sciences, University of Hull, Hull, HU6 7RX, UK
[5] IFP Energies Nouvelles, Geosciences Division, 1 et 4 Avenue de Bois-Préau, 92852 Rueil-Malmaison Cedex, France
[6] NERC Isotope Geosciences Laboratory, British Geological Survey, Nottingham NG12 5GG, UK and Centre for Environmental Geochemistry, School of Biosciences, University of Nottingam, Sutton Bonington Campus, Leicestershire, UK
[7] ICMB, Oldenburg University, P. O. Box 2503, 26111 Oldenburg, Germany
[8] Lyell Centre, Heriot-Watt University, Edinburgh, EH14 4AS, UK

## Abstract

The Kimmeridge Clay Formation (KCF) is a laterally extensive, total organic carbon-rich succession deposited throughout Northwest Europe during the Kimmeridgian–Tithonian (Late Jurassic). It has recently been postulated that an expanded Hadley Cell, with an intensified but alternating hydrological cycle, heavily influenced sedimentation and total organic carbon (TOC) enrichment, through promoting the primary productivity and organic matter burial, in the UK sectors of the Boreal Seaway. Consistent with such climate boundary conditions, petrographic observations, total organic carbon and carbonate contents, and major and trace element data presented here indicate that the KCF of the Cleveland Basin was deposited in the Laurasian Seaway under the influence of these conditions.

Depositional conditions alternated between three states that produced a distinct cyclicity in the lithological and geochemical records: lower variability mudstone intervals (LVMIs) which comprise of clay-rich mudstone, and higher variability mudstone intervals (HVMIs), which comprise TOC-rich sedimentation and carbonate-rich sedimentation. The lower variability mudstone intervals dominate the studied interval but are punctuated by three ~2-4 m thick intervals of alternating TOC-rich and carbonate-rich sedimentation (HVMIs). During the lower variability mudstone intervals, conditions were quiescent with oxic to sub-oxic bottom water conditions. During the higher variability mudstone intervals, highly dynamic conditions resulted in repeated switching of the redox system in a way similar to the modern deep basins of the Baltic Sea. During carbonate-rich sedimentation, oxic conditions prevailed, most likely due to elevated depositional energies at the seafloor by current/wave action. During TOC-rich sedimentation, intermittent anoxic-euxinic conditions led to an enrichment of redox sensitive/sulphide forming trace metals at the seafloor and a preservation of organic matter, and an active Mn-Fe particulate shuttle delivered redox sensitive/sulphide forming trace metals to the seafloor. In addition, based on

TOC-S-Fe relationships, organic matter sulphurisation appears to have increased organic material preservation in about half of the analysed samples throughout the core, while the remaining samples were either dominated by excess Fe input into the system or experienced pyrite oxidation and sulphur loss during oxygenation events. New Hg/TOC data do not provide evidence of increased volcanism during this time, consistent with previous work. Set in the context of recent climate modelling, our study provides a comprehensive example of the dynamic climate-driven depositional and redox conditions that can control TOC and metal accumulations in a shallow epicontinental sea, and is therefore key to understanding the formation of similar deposits throughout Earth's history.

## 1. Introduction

It is widely accepted that fine-grained marine sedimentary rocks preserve the most complete record of Earth's history (but see (Trabucho-Alexandre, 2015) and references therein for discussion). As such, understanding their formation is fundamental to investigating changes in climate, weathering regime, biogeochemical cycles, sedimentation, land–ocean linkages, and environmental change throughout geological time. Organic carbon-enriched mudstones are particularly pertinent to this as they represent dramatic perturbations in the carbon cycle, and might therefore be of value in predicting future environmental dynamics. Such mudstones also represent some of the most important global energy resources: petroleum source rocks. Understanding their formation is therefore of economic as well as environmental and broader scientific interest.

The Phanerozoic sedimentary record is punctuated by several instances of increased organic carbon burial, for example, the oceanic anoxic events during the early Toarcian, early Aptian, and early Albian (Jenkyns, 2010). The deposition of organic carbon-rich mudstones results from an interplay between the production and preservation of organic material as well as the dilution of organic material by other sedimentary components (e.g. Sageman et al., 2003). Sediment source and supply, primary production, biogeochemical cycling, water column redox conditions, and ocean connectivity have major implications for the formation of organic carbon-rich mudstones. However, the relative role of each of these processes is widely debated (Demaison and Moore, 1980; Pedersen and Calvert, 1990; Demaison et al., 1991; Tyson, 2001; Katz, 2005). The chemical makeup of a sedimentary succession can be used to assess these processes and reveal much about the depositional system, palaeoclimate, and basin history during deposition (e.g. Brumsack, 2006). When combined with petrographic observations, such analysis is a powerful tool with which environmental change can be determined over geological time.

A conceptual model developed for the Cretaceous (Hofmann and Wagner, 2011; Wagner et al., 2013) and further expanded to the Late Jurassic (Armstrong et al., 2016) linked the deposition of the total organic carbon-rich Kimmeridge Clay Formation (KCF) to atmospheric dynamics, specifically the shift and expansion of the orbitally modulated subtropical-tropical Hadley Cell, and associated changes in precipitation. Based on data from the type successions in Dorset, UK,

Armstrong et al. (2016) proposed that organic carbon-lean mudstones in the Kimmeridge Clay Formation were deposited during drier intervals, characterised by mixed-layer oxygenated conditions, whereas organic carbon-rich intervals were deposited under monsoonal conditions similar to those seen in the present-day tropics. During such intervals, enhanced fresh water and nutrient input from continental runoff produced stratified basins across the Boreal Seaway (Armstrong et al. 2016). The chronostratigraphic framework indicated that major fluctuations between wet and dry conditions occurred on a short eccentricity (100 kyr) timescale (Huang et al., 2010; Armstrong et al., 2016).

Here we present total organic carbon (TOC), total sulphur (TS), carbonate, major and trace element contents, and carbon isotope data, together with petrographic observations, for the Ebberston 87 Core (Fig. 1a) in order to investigate the primary controls on Late Jurassic sedimentation and TOC enrichment in the Cleveland Basin (Yorkshire, UK) and to further refine the hypotheses proposed by Armstrong et al. (2016).

## 2. Geological setting

Deposited during the Kimmeridgian and Tithonian stages of the Late Jurassic, the Kimmeridge Clay Formation (KCF) is up to 620 m thick and is biostratigraphically and geochemically correlated across Northwest Europe. In the Late Jurassic, atmospheric carbon dioxide concentrations were more than four times greater than those of the present day (Sellwood and Valdes, 2008). Much of northwest Europe was submerged by a shallow epicontinental sea, the Laurasian Seaway. The Laurasian Seaway comprised a series of interconnected basins, one of which was the Cleveland Basin, formed as a result of differential subsidence that started in the Triassic and continued through the Jurassic and Cretaceous (Rawson et al., 2000). The seaway connected the Boreal Sea to the Tethys Ocean through the Viking Corridor, situated between ~35 and ~40° N palaeo-latitude (Korte et al., 2015)(Fig. 1b).

Recent palaeoclimate modelling, paired with geochemistry and sedimentology from the Dorset and Yorkshire basins, indicates that the Laurasian Seaway was affected by a hot, humid climate comparable to present-day tropical monsoon regions (Armstrong et al., 2016). Deposition under these conditions was posited to have been strongly influenced by an intensified hydrological cycle and associated storm events. This would have promoted organic carbon enrichment through enhanced primary productivity from increased precipitation and continental runoff-drive nutrient supply, ocean overturn, and enhanced OM preservation from salinity/temperature stratification and reduction in bottom water ventilation. However, water depth reconstruction for the Laurasian Seaway are contentious and depend upon the preferred depositional model (Bradshaw et al., 1992). A water depth of tens of metres has been proposed by some authors (e.g. Hallam, 1975; Aigner, 1980; Oschmann, 1988), while others (e.g. Gallois, 1976; Haq et al., 1988; Herbin et al., 1991) have suggested a major Late Jurassic transgression that led to water depths of hundreds of metres. Reconstructions of sea-level change do not correlate with cycles in organic matter content indicating sea-level is not a primary control on organic matter enrichment (Herbin et al., 1993; Williams et al., 2001).

The KCF type section is exceptionally well-exposed along the coast around Kimmeridge Bay (Dorset, UK; Fig. 1b) and was cored as part of a high-resolution analysis project (see e.g. Morgans-Bell et al., 2001). It has been extensively studied by sedimentologists (Macquaker and Gawthorpe, 1993; Macquaker et al., 2010), stratigraphers (Morgans-Bell et al., 2001), palaeontologists (Lees et al., 2004), geochemists (Pearce et al., 2010), and palaeoclimatologists (Hesselbo et al., 2009) all trying to unravel the processes responsible for its deposition.

Owing to a lack of coastal outcrops, however, study of laterally equivalent deposits in the Cleveland Basin (Yorkshire, UK) has been limited to four cores drilled in the 1980s ( Herbin et al., 1991; Herbin et al., 1993; Herbin and Geyssant, 1993; Tribovillard et al., 1994; Boussafir et al., 1995; Herbin et al., 1995; Boussafir and Lallier-Vergès, 1997; Lallier-Vergès et al., 1997; Tribovillard et al., 2004). These four cores (Marton 87, Flixton 87, Ebberston 87, and Reighton 87; Fig. 1c) have been correlated to the type section in Dorset using the so-called 'Organic Rich Bands' (ORBs) (See Fig. 6 in Armstrong et al., 2016) defined by Gallois (1979) and later referred to by Herbin and Geyssant (1993).

The focus of the present study is the Ebberston 87 Core (54°12'47.7"N 0°37'22.7"W), which is thermally immature (average Tmax = 425 degrees C; Herbin et al. 1993), and contains type II and III kerogens (Herbin et al., 1993; Scotchman, 1991). The interval of detailed analysis spans the *Pectinatites wheatleyensis* to *Pectinatites pectinatus* ammonite biozones and is equivalent to the ORBs 4 and 5 of Herbin and Geyssant (1993) (Fig. 1a).

## 3. Materials and methods

A total of 116 samples were collected from the Ebberston 87 Core, stored in the IFP Energies Nouvelles (IFPEN) facilities (Chartres, France). Samples of 1 cm thickness were collected at 50 cm intervals through a 40-metre-thick section of the core. Sampling resolution was increased to 10 cm in darker, more organic carbon-rich intervals. Using established chronology of the Ebberston 87 Core (Herbin et al., 1993), and assuming a linear sedimentation rate, the studied interval spans approximately 800 kyr.

A total of 47 samples were prepared as thin sections and examined under optical light and a Scanning Electron Microscope (SEM). Samples were divided into microlithofacies based on composition, texture, and bedding features, following the nomenclature guidelines set out by Lazar et al. (2015), whereby sand is defined as any grain between 62.5 µm and 2000 µm, coarse mud is between 62.5 µm and 32 µm, medium mud is between 32 µm and 8 µm and fine mud is anything less than 8 µm. Grain size was measured using the rule in the SEM Aztec Software.

Total organic carbon (TOC) and total carbon (TC) were measured by LECO combustion analysis at Newcastle University. Equivalent $CaCO_3$ contents were calculated as $CaCO_3$ = (TC-TOC)* Molar Mass $CaCO_3$ / Molar Mass C = (TC-TOC)*8.33. Wavelength-Dispersive X-Ray Fluorescence (XRF) analyses were conducted at the Institute for Chemistry and Biology of the Marine Environment (ICBM, University of Oldenburg) to determine major (Si, Ti, Al, and Fe) and trace (Mn, As, Co, Cr, Cu, Mo, V, U, Zn and Zr) element contents of all samples. 114 samples were decalcified with 5 % HCl to remove the

carbonate and analysed for $\delta^{13}C_{org}$ values (calculated to the VPDB standard) using a Costech ECS4010 Elemental Analyzer connected to a VG TripleTrap and Optima dual-inlet mass spectrometer using within-run laboratory standards calibrated against NBS18, NBS-19 and NBS 22 at the BGS stable isotope facility (part of the National Environmental Isotope Facility). Precision of ± <0.1‰ (1 SD) was recorded. 49 samples were analysed for select trace elements (Mo, Cd, U, V, Re, Tl, As, and Sb) and Hg contents by Quadrupole Inductively Coupled Plasma Mass Spectrometry and a direct mercury analyser (DMA80, Milestone), respectively, at the Leibniz Institute for Baltic Sea Research (IOW). Sediments from the Landsort Deep (Baltic Sea) were analysed for Al and the above-mentioned trace metals by Inductively Coupled Plasma Optical Emission Spectrometry and ICP-MS, respectively. Major and trace elements are expressed as wt %, ppm, and ppb, as appropriate. See supplement A for details on sample preparation and analytical procedures.

Major and trace element contents were normalised to Al to allow for assessment of relative changes irrespective of dilution by organic matter or carbonate. Aluminium was chosen as representative of the siliciclastic fine-grained sediment fraction due to its generally high contents in the samples and its limited involvement in biological, redox and diagenetic processes (Tribovillard et al., 2006). Element/Al ratios are expressed as wt%/wt%, ppm/wt%, or ppb/% as appropriate. Trace element enrichment factors (EFs; Brumsack, 2006) were calculated relative to element/Al ratio of Upper Continental Crust (UCC; Rudnick and Gao, 2003).

**Linear sedimentation rates (LSRs) were calculated for the *P. wheatleyensis*, *P. hudlestoni*, and *P. pectinatus* biozones using ammonite zonal boundary dates in the Geologic Timescale (Gradstein et al. 2012). 4. Results**

Based on integrated geochemical data and petrographic observations, the studied section is divided into four lower variability mudstone intervals (LVMIs) and three distinct higher variability mudstone intervals (HVMIs) at 37–40 m, 45–47 m, and 65–69 m core depth (Fig. 2). The HVMIs are defined by distinct alternating facies along with repeated extreme enrichments of total organic carbon contents and trace elements, as shown on Figure 2. The *P. wheatleyensis*, *P. hudlestoni*, and *P. pectinatus* zones have LSRs of 6.6, 5.1, and 4.1 cm/ky, respectively.

**4.1 Petrographic characterization**

We defined six lithofacies based on compositional and textural observations (Fig. 3): 1) clastic detritus-rich medium-grained mudstone, 2) organic material and calcareous pellet-rich laminated medium to coarse mudstone, 3) coccolith-dominated medium mudstone, 4) agglutinated foraminifera-bearing, medium to coarse carbonaceous mudstone, 5) biogenic detritus-dominated fine to medium mudstone, and 6) carbonate-cemented coarse mudstone. The lower variability mudstone intervals are comprised of facies 1, 5 and 6, while the higher variability mudstone intervals are dominated by facies 2, 3, and 4

### 4.1.1 Lower variability mudstone interval (LVMI) petrography

Facies 1 comprises an argillaceous matrix containing medium to coarse mud-sized grains (Fig. 3). The coarsest grain fraction is fine to coarse mud-sized, unrounded quartz grains, occasional medium mud-sized chlorite grains, fine mud-sized titanium oxide grains, and very occasional phosphatic clasts. Equant pieces of organic matter (2–5 µm in diameter) and framboidal pyrite are finely disseminated throughout the matrix. Occasional organo-minerallic aggregates, comprised of wispy orange organic material (OM) and clay material, are found within the matrix. Calcitic microfossils, predominantly foraminifera tests and disarticulated shell fragments, are present, and commonly infilled with authigenic kaolinite. These mudstones have a churned texture so most of the primary sedimentary structures have been destroyed; however, some samples contain infilled burrows indicating the sediment was bioturbated (Fig. 4a). Physical mixing processes, indicated by occasional erosional surfaces, probably contributed the homogenisation of the sediment (Fig. 4b).

Facies 4 is very similar to Facies 1 but contains sand-sized quartz and is dominated by disarticulated calcitic shells, some of which are wholly or partially replaced by pyrite (Fig. 3).

Facies 6 is a medium to coarse mud-sized, angular, diagenetic carbonate grain-dominated sedimentary rock with an argillaceous matrix. Carbonate grains are microcrystalline, zoned, non-ferroan calcite and non-ferroan dolomite crystals. Relicts of the primary matrix, comprised of coccolith debris, organic matter, illitic clay and fine quartz grains, can be seen between the diagenetic components. The three samples representing this facies type (Fig. 2, three arrows in $CaCO_3$ plot) are not included in the palaeoenvironmental interpretation as their primary sedimentary and geochemical signature appears to be strongly overprinted by diagenesis.

### 4.1.2 Higher variability mudstone interval (HVMI) petrography

Facies 2 comprises sub-mm length, discontinuous, wavy lamina organized into normally graded centimetre scale beds with erosional bases. The facies is dominated by amorphous algal marine-derived organic material aligning with and wrapped around the other sedimentary components, including the clay matrix, quartz grains, microfossils, lithic clasts, coccoliths and framboidal pyrite (Fig. 3). The algal maceral commonly exist within organo-minerallic aggregates, and extensive 'network' structures that are 500 µm thick and up to 2mm in length point towards algal mat deposition (Fig. 4c,d). Coccoliths commonly occur within faecal pellets but also in the matrix (yellow arrows in Fig. 4c,d).

Facies 3 comprises normally-graded, sub-mm laminae with erosional bases. The predominant sedimentary components are coccolith-rich faecal pellets that range in size from 30–3000 µm, unlike Facies 2 where Type II OM is a main sedimentary component (Fig. 3). The matrix is predominately pristine coccoliths and coccospheres but also contains clay minerals, equant lumps of organic material, and wispy algal macerals, calcispheres, chlorite grains, coarse mud-sized sub-rounded quartz grains, pyrite framboids (Fig. 4e), euhedral pyrite crystals (Fig. 4e), occasional agglutinated foraminifera (Fig. 3), some of which are infilled with authigenic kaolinite (Fig. 4f).

Facies 5 is similar to Facies 2 in that it has a carbonaceous, calcareous, and argillaceous matrix with medium to coarse mud-size grains. However, abundant agglutinated foraminifera, composed of quartz grains, clay minerals, and pyrite crystals, are observed in this facies. It is homogenised due to a combination of extensive bioturbation indicated by burrows and physical mixing suggested by occasional relict lamina.

## 4.2 Geochemistry

Figure 5 shows a ternary diagram with the main inorganic lithogenic components represented by key proxy elements (Brumsack, 1989): clay is denoted by $Al_2O_3$, quartz by $SiO_2$, and carbonate by CaO. All samples fall on or very close to a mixing line between the calcium carbonate end member and a quartz-clay mixture that is more clay-rich than average shale (Wedepohl, 1971; Wedepohl, 1991) but less clay-rich than Upper Continental Crust (Rudnick and Gao, 2003).

Across the studied core section, total organic carbon (TOC) contents range from 0.8 wt % to 21.8 wt % with a mean of 6.6 wt %. Equivalent $CaCO_3$ contents range between 0.5 wt % and 73.8 wt % with a mean value of 28.5 wt % (Fig. 2). The $CaCO_3$ contents of three samples (51.0 m, 53.3 m, and 57.0 m; indicated by arrows in Fig. 2) are likely to be erroneous given the abundance of dolomite identified in the petrographic analysis (Fig. 3). Petrographic and geochemical characteristics indicate strong overprinting by later-stage diagenesis (Facies 6; Fig. 3) so these samples are discounted from palaeoenvironmental interpretation.

### 4.2.1 Lower variability mudstone interval (LVMI) geochemistry

The lower variability mudstone intervals are characterised by TOC contents ranging from 0.8 wt % to 10.4 wt % (Fig. 2) with a mean of 4.0 wt %. $\delta^{13}C_{org}$ ranges from –25.7 ‰ to –27.9 ‰ with a mean of –26.7 ‰ (Fig. 2).

Silicon and Al range between 6.1 and 22.2 wt %, and between 3.1 and 10.4 wt %, respectively. The Al-normalised ratios of Si, Ti, and Zr (indicative of coarse grain sizes and/or heavy minerals; Dellwig et al., 2000) range from 2.0 to 2.6, 0.04 to 0.05, and 9.8 to 17.6, respectively (Fig. 6). All redox sensitive/sulphide forming trace metals (Mo, U, V, Re, Tl, As, Sb, Hg, Fe, and Mn; Fig. 7, Cu, Zn, Cd; Fig. B1) are consistently lower in the LVMIs compared to the HVMIs. Enrichment factors (Fig. C1) calculated relative to UCC for Mo, Re, Tl, As, Sb, Hg, Cu, Zn, and Cd are generally around 1 for the LVMIs. Uranium and V are slightly greater than 1 indicating a slight enrichment relative to UCC.

### 4.2.2 Higher variability mudstone interval (HVMI) geochemistry

In the three higher variability mudstone intervals (HVMIs), TOC ranges from 2.8 wt % to 25.7 wt % with a mean of 11.7 wt %. Calcium carbonate contents range from 6.8 wt % to 88.3 wt %, and a mean of 36.0 wt %. $\delta^{13}C_{org}$ ranges from –27.0 ‰ to –21.8 ‰, and the average of –25.3 ‰ is higher than in the lower variability mudstone intervals (Fig. 2).

The records of Si and Al exhibit larger ranges in the HVMIs than in the LVMIs and range from 1.9 to 23.9 wt %, and from 0.59 to 10.3 wt %, respectively. The coarse grain size/heavy mineral bound elements are slightly higher than in the LVMIs.

Si/Al, Ti/Al, and Zr/Al range from 2.0 to 3.3, 0.04 to 0.06, and 10.6 to 26.7, respectively (Fig. 6). Silica is positively correlated with Al ($R^2 = 0.98$), which suggests that the Si is strongly associated with the clay fraction rather than biogenic, volcanogenic or diagenetic silica in the sediment, consistent with petrographic observations. We can therefore use Si/Al in conjunction with Ti/Al and Zr/Al ratios as proxies for variations in grain size, and thus depositional energy (Tribovillard et al., 2006). The grain size indicators presented here (Fig. 6) suggest the HVMIs are slightly coarser-grained and more variable than the LVMIs. However, for samples with Al value close to 0, this could partly be an artefact of normalisation to Al, as some values in the HVMIs are low in Al contents compared to the LVMIs, probably due to the increased dilution of background terrigenous input by organic material and biogenic carbonate (Fig. 6). Therefore, normalisation to Al in these intervals may produce spuriously high grain size proxy results and should be regarded with caution (For further discussion see Van der Weijden, 2002).

All redox-sensitive/sulphide-forming trace metals (Mo, U, V, Re, Tl, As, Sb, Hg, Fe, and Mn; Fig. 7, Cu, Zn, Cd; Fig. B1) are variable and enriched relative to the LVMIs. Notably, Mn/Al correlates with $CaCO_3$ content for the HVMIs (Fig. C1). Enrichment factors (Fig. D1) calculated relative to UCC for Mo, U, V, Re, Tl, As, Sb, Cu, Zn, and Cd are variable but generally > 1 in the HVMIs, indicating their enrichment relative to UCC.

## 5. Discussion

### 5.1 Productivity and organic matter composition

Biological components (coccolithophores, foraminiferans, and organic matter (OM)) occur in differing proportions throughout the section (Figs. 2 and 3). Our petrographic observations (Figs. 3 and 4) and organic carbon isotope data (Fig. 2) indicate a mixed OM type II-III component, in agreement with the published RockEval data that determines a Type II-III OM source (Herbin et al., 1993; Scotchman, 1991). Type II OM constitutes marine OM such as algal macerals, and has a higher $\delta^{13}C_{org}$ than Type III OM, which is terrigenous (Leng and Lewis, 2017). Fluctuations in $\delta^{13}C_{org}$ correspond to the changes in dominant OM source (marine versus terrestrial) as demonstrated in the petrography; therefore, we use $\delta^{13}C_{org}$ as a proxy for OM source.

### 5.1.1 Productivity and organic matter composition in the lower variability mudstone intervals (LVMIs)

Type III OM, with typical carbon isotopes values of –26 to –28 ‰ VPDB, is ubiquitous in the LVMIs (Figs. 2, 3, and 4). The OM was produced on adjacent land masses and was transported into the marine environment along with other detrital components, such as quartz and clay minerals. The largely very fine-grained nature of the LVMIs suggests that organic material, together with clay-sized siliciclastic detritus, was preferentially deposited away from a major coarse-grained river system. Furthermore, Type III organic material is more resistant to oxidative degradation and hydrodynamic disaggregation due to its structure and chemistry, which explains how organic material enrichment in the LVMIs (average TOC = 4.0 wt %)

is possible in an environment that had an active benthic ecosystem indicative of oxic-suboxic bottom water conditions. In addition to abundant terrestrial organic material, there are occasional algal macerals in the LVMIs.

**5.1.2 Productivity and organic matter composition in the higher variability mudstone intervals**

In contrast to the LVMIs, the TOC-rich intervals of the HVMIs (average TOC = 12.3 wt %) are dominated by Type II OM. Wispy lamina, algal maceral material (Figs. 3 and 4), and algal mats are the main source of OM in these intervals, although Type III OM remains ubiquitous and coincides with the occurrence of detrital material, i.e. quartz grains, lithic clasts, and clay minerals in the samples. In the TOC-rich parts of the HVMIs, large amounts of type II algal derived material deposited at the seafloor acted to dilute the detrital sediment and type III OM.

The occurrence of organo-minerallic aggregates and algal OM (20 μm thick and 200 μm long; Fig. 3) demonstrate marine snow and algal mat settling as key mechanisms for the delivery of OM to the seafloor (Macquaker et al., 2010). Filter-feeding organisms strip nutrients and fine grained sediment out of the water column and excrete them as faecal pellets (Ittekkot et al., 1992). This biological mediation of the sediment is a key process in the export of OM, sediment, and nutrients from the water column to the sediment. Ittekkot et al. (1992) demonstrated a link between enhanced sediment and nutrient flux to the ocean during wet phases of the monsoonal cycle. They showed also that there was an increase in biotic-abiotic pellet production, and thus an enhanced OM and mineral flux to the seafloor, in response to enhanced continental runoff and weathering associated with the monsoonal wet phases.

Production of algal macerals in the oceans is affected by the availability of light and nutrients (Pedersen and Calvert, 1990). Given the palaeolatitudinal positioning of the section (Fig. 1b) it is likely that light did not limit primary productivity. We therefore propose that nutrient availability was the driver of changes in productivity. Nutrients are supplied to the euphotic zone through continental weathering or upwelling/overturning of nutrient rich waters. Given the palaeogeographic setting of the Cleveland Basin in an epicontinental seaway, it is likely that changes in precipitation on the surrounding landmasses were a key driver of changes in productivity. The wet-dry cycles proposed by recent climate modelling (Armstrong et al., 2016) may therefore be the key driver behind oscillations in the production and preservation of TOC, i.e. the switching between the LVMIs and HVMIs. Alternatively, the variations in TOC could result from variable dilution of the OM by carbonate or detrital material, or the variations could result from a combination of both dilution and wet-dry cycles.

Carbonate productivity, mainly in the form of coccoliths, varies throughout the studied KCF section and is at its maximum within the carbonate-rich sections of the HVMIs. High nutrient availability in the photic zone of the water column, indicated by high OM productivity, also influences the rate of carbonate productivity (Lees et al., 2006). Within the HVMI's, alternations between TOC-rich deposition likely occurred during high nutrient availability, and carbonate-rich deposition is more likely under less nutrient-rich conditions. This alternation may result from ecological switching driven by oceanographic or climate processes that subtly altered the nutrient levels during these intervals. The organisation of the coccoliths into faecal pellets in both the TOC-rich and carbonate-rich intervals of the HVMIs indicates a high abundance of

higher trophic organisms to graze on a plentiful supply of food further supporting an elevated nutrient level during these intervals. Furthermore, feacal pellets reduce the time taken for sediment to reach the seafloor (e.g. Shanks, 2002) so the abundance of pellets and organo-minerallic aggregates led to increased OM export efficiency thus enhanced OM preservation (Macquaker et al., 2010). For the Dorset section, Macquaker et al. (2010) suggested that zooplankton were the

main grazers and producers of faecal pellets. Our petrographic study of the Ebberston 87 Core reveals strong similarities with the type section in Dorset, so we tentatively suggest that zooplankton were also the main grazers in the Cleveland Basin at the time of deposition. Alternatively, variations in carbonate may be as a result of dilution by other sedimentary components.

## 5.2 Depositional environment: sediment source, depositional energy, dispersal mechanisms, and climatic context

The studied succession is a three-component system, comprising clay-dominated, organic carbon-dominated, and carbonate-dominated facies (Fig. 3). The clay fraction of the mudstones is probably derived from detrital inputs to the basin, through the weathering of nearby emergent landmasses. Possible sources include Cornubia, or the Welsh, Irish, or London-Brabant landmasses (Fig. 1b), as discussed by Hesselbo et al. (2009). Terrestrial organic matter (OM) was likely washed into the basin along with detrital clays, while carbonate and marine OM formed in the water column.

On geological timescales, mercury is released in large volumes during volcanism (Percival et al., 2015). A relatively long atmospheric residence time (1-2 years) means it can be globally distributed prior to deposition in the sedimentary record. Mercury enters the ocean primarily through precipitation, once in the ocean it is preferentially adsorbed on to organic matter. Assuming the Hg is adsorbed to the organic material, assessment of Hg/TOC ratios can indicate an increased/reduced supply of Hg in the system (Percival et al., 2015), hence it can be utilised as a proxy for volcanism. However, caution must be taken

when interpreting records of Hg enrichments in sedimentary rocks because Hg can accumulate as a result of several other processes (e.g. Them et al., 2019; Grasby et al., 2019). Diagenesis may affect Hg contents, depending upon which sedimentary component hosts the it (e.g. organic matter or pyrite), ongoing research is still exploring this (Them et al., 2019). A volcanogenic sediment source is ruled out using Hg/TOC as a proxy (Scaife et al., 2017)(Fig. 6), which agrees with other studies of the KCF (Percival et al., 2015), and petrographic evidence (conchoidally fractured quartz grains without Fe

oxide rims) rules out an aeolian source, which is also supported by the palaeogeographical position of this basin in the outer palaeo-subtropics (Armstrong et al., 2016).

The $SiO_2$, $Al_2O_3$, and $CaCO_3$ ternary diagram (Fig. 5) suggests the studied mudstones have an overall finer grain size than average shale. This may result from either the presence of more clay material or a higher contribution of Al-rich clay minerals (e.g., kaolinite), however petrographic results show a predominance of illite (K-Al rich clay mineral) in the matrix

suggesting it is more clay-rich than average shale, which corroborates published XRD analyses (Herbin et al., 1991). It is possible that either the coarse-grained siliciclastic fraction of the sediment was trapped in nearshore proximal settings, with only fine-grained material being transported to more distal settings within the Cleveland Basin, or that the fine-grained

nature of the sediment may result from the weathering of fine-grained source material on the hinterland or the deposition of palimpsest sediment that was deposited and remobilised on the ocean. Nevertheless, linear sedimentation rates (LSRs) were calculated of 6.6, 5.1, and 4.1 cm/ky for the *P. wheatleyensis*, *P. hudlestoni*, and *P. pectinatus* ammonite biozones, respectively, indicate the system was not starved of sediment.

Depth plots of geochemical grain size proxies (Fig. 6) indicate generally quiescent conditions in the LVMIs, with slightly more variable energy in the HVMIs. Petrographic observations support the geochemical grain size proxies, showing a lower proportion of clay material and a dominance of alternating organic carbon- and carbonate-rich sediment in the HVMIs (particularly in facies 2 and 3). In addition to the compositional variations, the occurrence of normally-graded beds with erosional bases in TOC-rich sections of the HVMIs indicates an energetically dynamic setting. Occasional higher energy

conditions affected sediment deposition and dispersal at the seafloor and led to winnowing of the finest grain sizes. Significant quantities of terrestrial OM (identified by $\delta^{13}C_{org}$ and petrography), coupled with its overall fine-grained nature, suggest that the sediments accumulated in a depositional setting where hydrodynamic processes sorted the sediment based on particle size and density.

Owing to the shallow gradients and vast extent of epicontinental seaways, sediment dispersal in the LVMIs, which are

dominated by terrigenous mud, may have been controlled by wind- and tide-induced bottom currents (Schieber, 2016); however, uncertainty on water depths in the Cleveland Basin at the time of deposition means it is not possible to determine the significance of these processes.

The occurrence of faecal pellets, increases in grain size, TOC, and $CaCO_3$ observed in the HVMIs studied here points towards enhanced productivity, which is likely to be a regional phenomenon. If this was not limited to the Cleveland Basin,

it may explain a coeval organic enrichment across the Laurasian Seaway. The causal mechanism behind the increase in productivity cannot be constrained with the present dataset. However, slight increases in depositional energy may result from intermittent storm mixing or changes in bottom water currents, which could both increase primary productivity.

Armstrong et al. (2016) attributed alternations in TOC enrichment to orbitally forced changes in humidity–aridity occurring over a short eccentricity timescale (100 kyr). Based on the chronostratigraphic time frame for the Yorkshire and Dorset

sections (Armstrong et al., 2016; Huang et al., 2010) and assuming a linear sedimentation rate, the changes between the HVMIs and LVMIs equate with short eccentricity cycles. Thus our observations match with the prediction of Armstrong et al. (2016) that enhanced TOC burial occurred during wet periods through enhanced continental weathering, nutrient supply to the ocean, and primary productivity. The astrochronological timescale established for the Swanworth Quarry 1 Core in the Wessex Basin (Huang et al., 2010) and applied to the Ebberston 87 Core in the Cleveland Basin (Armstrong et al., 2016) is

plotted for references in Fig.1a. Given the relatively short duration of the studied interval, it is not possible to conduct further spectral analysis with the present dataset.

### 5.3 Redox conditions

#### 5.3.1 Redox conditions in the lower variability mudstone intervals (LVMIs)

In the lower variability mudstone intervals, petrographic results demonstrate extensive bioturbation, most likely resulting from widespread faunal colonisation of the sediment surface due to well-oxygenated bottom waters (Fig. 4). However, enrichment factors of redox-sensitive elements (Fig. 7) (Brumsack, 2006; Tribovillard et al., 2006; Piper and Calvert, 2009) indicate that during the deposition of the LVMIs, the sediment pore water was suboxic. This interpretation is further supported by the low Mn/Al and high (TOC=0.8–10.4 wt %; Fig. 2) TOC contents.

It is difficult to reconcile the high TOC contents, which are considered extremely TOC-rich and qualifies as a potential hydrocarbon source rock, with oxygenated bottom waters. We explain this apparent conundrum by the presence of terrestrial OM in the LVMIs as demonstrated by lower $\delta^{13}C_{org}$ (Fig. 2), petrography (Fig. 3) from this study, and from published Rock Eval data (Herbin et al., 1993). In comparison to its marine counterpart, terrestrial organic matter often has a higher preservation potential because its chemical structure can make it less susceptible to oxidative degradation and because sulphate reducing bacteria have a lower affinity for terrestrial OM (Dellwig et al., 2001). Similarly, marine algal macerals have a greater chance of being preserved when incorporated into organo-minerallic aggregates that are observed in the HVMIs, because physical and chemical attractions between the aggregate components form a protective barrier to oxidants thus reduce oxidative destruction of the algal macerals (Macquaker et al., 2010).

#### 5.3.2 Redox conditions in the higher variability mudstone intervals (HVMIs)

The TOC-rich parts of the HVMIs are dominated by algal macerals (Figs. 3 and 4), which account for the elevated TOC concentrations of up to 25.7 wt % (Fig. 2). Enrichments in redox sensitive/sulphide forming elements (Mo, U, V, Zn, Cd, Re, As, Sb, Hg, and Fe; Fig. 7, Zn and Cd; Fig. B1) clearly support the notion of periodically anoxic to euxinic pore and bottom waters at the study site (Brumsack, 2006; Tribovillard et al., 2006), which enhanced preservation of OM during these intervals. However, the presence of agglutinated foraminifera within the TOC-rich parts of the HVMIs, as well as enrichments of Mn relative to the LVMIs (Fig. 7), argue that oxygen must have been present at the seafloor, at least episodically (Dellwig et al., 2018; Macquaker et al., 2010). A positive correlation between Mn/Al and $CaCO_3$ in the HVMIs (Fig. B1) is further evidence that during favourable conditions for $CaCO_3$ formation, Mn was sequestered into the sediment thus implying repeated oxygenation events (but see Herndon et al., 2018) for alternative discussion). Nevertheless, this scenario has similarities to the modern Baltic Sea deeps (Landsort Deep, Gotland Basin) that display dramatic shifts between euxinic conditions during stagnation and oxic conditions during North Sea water incursions, leading to the sequestration of large amounts of Mn carbonate into the sediments (Dellwig et al., 2018; Häusler et al., 2018; Scholz et al., 2018).

### 5.3.3 Fe enrichment and sulphide formation

The significant enrichments of Fe in the TOC-rich intervals of the HVMIs are related to microbial sulphate reduction and pyrite formation, following dissolution of Fe (oxyhydr)oxides at the seafloor. The formation of pyrite ($FeS_2$) depends on the redox conditions of the sediment pore and overlying water column, so it can be used to reconstruct palaeoenvironmental conditions beyond the redox state (Hetzel et al., 2011). Figure 8 shows a $Fe_x$-TOC-S ternary diagram for the studied interval. Reactive iron is calculated as $Fe_x = Fe - 0.25*Al$ to account for the fraction of Fe that is bound in the silicate fraction and unavailable for redox reactions (Brumsack, 1988; Dellwig et al., 1999). A fully quantified measurement of reactive iron in a sample must be determined experimentally, owing to an absence of this, we discuss samples only in a relative way. For the samples plotting above the TOC-pyrite mixing line, pyrite formation was limited by the availability of sulphur (in the form of $H_2S$ generated by bacterial sulphate reduction) and excess, less-pyritised reactive iron was preserved in the sediments (potentially as oxides or carbonates). Both in the LVMIs and the HVMIs, there are samples with an excess of reactive Fe relative to S. Limited availability of reactive OM to support bacterial sulphate reduction (i.e. limited $H_2S$ generation) may be the reason for the observed excess iron. Alternatively, it might either be related to the re-oxidation of pyrite during oxygenation events (removing S but not Fe from the sediment), or a strong input of Fe into the system via a particulate shuttle mechanism (discussed in sect. 5.3.4).

The samples that plot near the TOC-pyrite mixing line in Figure 8 are assumed to represent an availability of reactive Fe and S in the system that matches the stoichiometry of pyrite, hence they have a higher degree of pyritisation. Pyritisation explains the enrichment of metals known to accumulate with pyrite (e.g., As, Sb and Mo; Tribovillard et al., 2004). In samples that plot below the TOC-pyrite mixing line in Figure 8, pyrite formation is limited by the availability of reactive iron, meaning the degree of pyritisation was high but there also was an excess of sulphide available in the system. Under these conditions, sulphide was able to react with OM through sulphurisation (or natural vulcanisation). Our results are consistent with (Tribovillard et al., 2004) who concluded that pyrite formation prior to OM sulphurisation may increase the sequestration of Mo into the sediment. This highlights the need for caution when using Mo as a single diagnostic proxy in palaeoenvironmental reconstruction. (Tribovillard et al., 2004) further demonstrated a linear relationship between the quantity of 'orange algal macerals' and Mo concentrations in the Ebberston 87 Core, supporting the presence of sulphurised OM. The resulting sulphurised OM is less vulnerable to oxidative degradation and thus has a greater preservation potential during burial. We suggest that this mechanism is partly the reason we observe high TOC contents in intervals that underwent first sustained periods of pore or bottom water euxinia and later periods of reoxygenation. Studying Jurassic sediments of the Marton 87 Core (Fig. 1c), drilled 4 km away from the Ebberston 87 Core, (Boussafir et al., 1995) and (Lallier-Vergès et al., 1997) also conclude that sulphurisation/natural vulcanisation of Type II OM enhanced organic material preservation potential.

### 5.3.4 Evidence for a Fe-Mn shuttle and ocean restriction

In the context of pyrite formation versus OM sulphurisation, and the enrichment of trace metals at the seafloor, the delivery of Fe and Mn (oxyhydr)oxides via a so-called particulate shuttle may play an important role. The particulate shuttle effect describes the cycling of Mn and Fe through the redoxcline, which is an ocean layer characterised by steep geochemical gradients, that are driven by changes, and the point at which solubility of certain mineral phases increases or diminishes, in particular Mn, Fe, and P phases (Dellwig et al., 2010). Dissolved $Mn^{2+}$ diffuses from anoxic waters beneath the redoxcline to the oxygenated surface layer where is it oxidised to form $MnO_2$ particles. These particles react with $Fe^{2+}$ to form mixed Mn/Fe (oxyhydr)oxide particulates which are highly efficient adsorbents of dissolved trace metals in the ocean (Goldberg, 1954). If these particulates sink and reach deeper sulphidic waters, the (oxyhydr)oxides are reduced, and $Mn^{2+}$ diffuses back into the overlying waters, while $Fe^{2+}$ reacts with $H_2S$ to form Fe sulphides that incorporate part of the released trace metals (Canfield et al., 1992; Huckriede and Meischner, 1996; Neretin et al., 2004; Dellwig et al., 2010; Tribovillard et al., 2015). This particulate shuttle plays a key role in the transfer of trace elements from the water column to the sediment (Goldberg, 1954; Tribovillard et al., 2015), and is particularly effective when redox conditions fluctuate (Algeo and Tribovillard, 2009). Of particular relevance to this study is the transfer of Mo, As, and Sb to the sediment, elements that Tribovillard et al. (2015) used as proxies for the presence of a Mn-Fe shuttle in ancient anoxic-euxinic ocean basins. In the present study, Fe/Al (Fig. 7) and Mo/U (Fig. 9a) co-variations point towards a Mn-Fe shuttle operating during deposition of TOC-rich sediment in the HVMIs. Manganese is repeatedly reduced and re-oxidised through the shuttle and does not get removed into the sediments as long as bottom water conditions are suboxic, leading to a build-up of dissolved Mn in the water column. Therefore, the Mn contents encountered in the oxic layers are easily accounted for due to fast deposition of Mn oxide from the water column and subsequent transformation into Mn carbonates within the sediments. Peaks in Fe/Al coincide with peaks in pyrite associated trace metals (As/Al, Sb/Al; Fig. 7) and increased abundance of pyrite in thin sections (Fig. 3). We propose that a suboxic shuttle that remobilised Fe from shelf sediments (Wijsman et al., 2001, Severmann et al., 2008, Dellwig et al., 2010; Eckert et al., 2013;) was active during deposition of the TOC-rich parts of the HVMIs in the Ebberston 87 Core, and enriched reactive Fe in the sediments to levels significantly above UCC. Episodically high inputs of reactive Fe via a suboxic shuttle also prevented the accumulation of $H_2S$ in the pore waters, thereby limiting OM sulphurisation. During ventilation events (as suggested by benthic foraminifera and Mn enrichments), some of the pyrite was reoxidised, and sulphate diffused from the pore water to the water column while reactive Fe was retained in the sediment by precipitation of Fe (oxyhydr)oxides. Thus, samples that appear to be deposited under S limited conditions in Figure 8 may still contain organic sulphur. We conclude that the hyper-accumulation of TOC in this setting was controlled by high primary productivity, increased OM production, the balance of $H_2S$ generation, and the supply of reactive Fe via a suboxic shuttle.

In the context of developing an anoxic/euxinic water column in the Cleveland Basin, not only primary productivity but also basin restriction has to be considered as a contributing factor. Due to the specific biogeochemical behaviour of Mo, Mo/U enrichment factors have been shown to give insights into the degree of restriction of a basin with examples from the Carico

Basin, the Black Sea, the Late Pennsylvanian Midcontinent Sea, and the Late Devonian Seaway (Algeo and Tribovillard, 2009). Both elements tend to be enriched in sediments deposited under oxygen-depleted bottom water conditions, but to varying degrees (Helz et al., 1996). Under suboxic conditions, U enrichment is likely to exceed Mo enrichment owing to the trapping of U at the Fe(II)/Fe(III) redox boundary (Algeo and Tribovillard, 2009). As oxygenation of the water mass decreases further, and sulphidic conditions develop in pore and bottom waters, Mo enrichment increases relative to that of U, and the Mo/U ratio approaches or surpasses that of seawater due to the presence of sufficient $H_2S$ to convert the conservative molybdate ion to the highly particle-reactive thiomolybdate (Algeo and Tribovillard, 2009; Erickson and Helz, 2000). Mo/U ratios for the LVMIs in this study are similar to that of average shale (Fig. 9a), pointing towards a detrital, rather than authigenic, source of the Mo and U in these samples. Therefore, these samples cannot be used to investigate the presence of a Mn-Fe shuttle. In contrast, the HVMIs display repeated peaks in the Mo/U ratio, indicative of strong Mo enrichments under euxinic conditions, facilitated by the activity of a Mn-Fe shuttle preferentially transferring Mo from the water column to the sediment (Algeo and Tribovillard, 2009).

We compare our new results with existing data from Tribovillard et al. (2004), who analysed trace elements (Fig. 9b) in several samples from the Ebberston 87 Core, with cycle 3 in their study being equivalent to the stratigraphically oldest (*Pectinatites wheatleyensis*) HVMI in our study. The authors concluded that TOC enrichment in the Cleveland Basin occurred during times of transgression due to siliciclastic sediment starvation, high primary productivity, and water column stratification. Our higher resolution sampled data agree with this conclusion in that on a Mo-U enrichment factor plot (Fig. 9b), the TOC-rich intervals plot clearly within the particulate shuttle field. Samples from the LVMIs plot outside of this field, which confirms that during deposition of these samples the suboxic conditions were limited to the sediment. Pearce et al. (2010) concluded that the deposition of the KCF occurred in an unrestricted basin. (Tribovillard et al., 2012) compared their results to those presented by Pearce et al. (2010) and agreed that is it unlikely that either the Cleveland or Wessex Basins were restricted for a prolonged period, if at all. In combination with our geochemical and petrographic data, we can confirm that the repeated development of anoxic/euxinic conditions in the Cleveland Basin was most likely due to high primary productivity, and possibly salinity stratification due to high amounts of freshwater runoff, but it is unlikely that the Cleveland Basin experienced prolonged restriction.

### 5.3.5 Comparison to modern organic carbon-rich sediments

Comparing our data with those from different TOC-rich deposits enables us to better understand common processes involved in TOC enrichment in different palaeoenvironmental/depositional settings and draw wider conclusions for the present study. Figure 10 shows average enrichment factors (calculated with respect to average shale for select trace metals, Mn, As, Cd, Co, Cu, Mo, Re, Sb, U, V, and Zn, to facilitate direct comparison with the data in Brumsack, 2006) for the carbonate-rich and TOC-rich sections of the HVMIs in the present study and other well-studied deposits. Depositional environments, in which TOC and trace elements are enriched, can fall on a scale between two end members; coastal upwelling systems (e.g.

Gulf of California, Peru coastal margin) and anoxic basin type settings (e.g. Mediterranean Sapropels, Black Sea, Baltic Sea deeps). The former are characterised by seasonally high primary productivity and extensive oxygen minimum zones (OMZ) that are fuelled by wind-driven upwelling of high nutrient and trace metal waters, while the latter are characterised by permanent salinity stratification and water mass restriction (Brumsack, 2006).

The HVMIs in the present study bear similarities to the Gulf of California in that they exhibit similarities in Cd enrichment and depleted in Mn in comparison to the HVMIs, indicating high productivity and suboxic conditions in the water column (Sweere et al., 2016). Cd enrichments suggest that parts of the HVMIs experienced upwelling like processes, which may have resulted in high productivity. However, sulphidic conditions in the water column are rarely observed in upwelling settings (Brumsack, 2006); this differs from the studied Cleveland Basin and is reflected in the sulphide indicators (As and Mo; Fig. 7). This indicates that the method of trace metal sequestration is different in modern upwelling settings and the studied section, meaning they are a poor analogue for the present study.

The anoxic basin type settings exhibit strong enrichments in sulphide forming trace metals (Brumsack, 2006). However, the enrichment factors for the Mediterranean Sapropels and Units 1 and 2 from the Black Sea exhibit different patterns (Fig. 10), owing to the pronounced depletion of trace elements in the water column of the Black Sea during Unit 1 and less pronounced euxinia during most parts of Unit 2 (Wegwerth et al., 2018). The studied HVMIs are most similar to Black Sea Unit 2 in that redox sensitive (As, Cu, Mo, Re, U, V, and Zn) and productivity (Cd) elements are on the same order of magnitude. From this, we may infer similarities operating in Black Sea Unit II and the HVMIs. However, the modern Black Sea is permanently stratified and is governed by its unique hydrological setting. This is in stark contrast to the HVMIs in the Cleveland Basin, which exhibit rapid oscillations and were deposited in a shallow epicontinental seaway. It may be that the Late Jurassic Laurasian Seaway fluctuated between a setting close to an anoxic basin type setting and setting more analogous to a coastal upwelling zone.

### 5.3.6 Evidence for Baltic Sea characteristics

While the studied interval shares similarities and differences with both upwelling and anoxic basin type settings, we are still lacking an appropriate modern analogue. Palaeogeography exerts a fundamental control on sedimentation, in particular, TOC enrichment, but there is no modern-day example of a shallow epicontinental seaway. See Algeo et al. (2008) for discussion of possible analogues. The Baltic Sea comprises a series of small basins that are interconnected by narrow sills and channels and bears resemblance to the Late Jurassic Laurasian Seaway. We propose that redox dynamics in the Cleveland Basin during the Late Jurassic shared similarities to those in the modern-day Baltic Sea in that they both fluctuated between LVMI and HVMI deposition. The enrichment factors in Fig. 10 are averaged across the HVMIs, indicating Mn depletion across these zones. However, there are several samples that are enriched in Mn within the HVMIs (Fig. 7), which point towards prolonged reoxygenation events during episodes of organic enrichment in the Kimmeridge Clay Formation.

The Baltic Sea receives saline water input from the North Sea and fresh water from riverine input resulting in permanent stratification of the water column (Dellwig et al., 2018; Häusler et al., 2018). Coupled restriction of water mass exchange in the deeper sub-basins of the Baltic Sea leads to appearance of water column anoxia/euxinia. However, large (and infrequent) inputs of saline and oxic water from the North Sea promote overturning and reoxygenation of the Baltic Sea deeps, causing
extensive Mn carbonate formation in the sediments (Dellwig et al., 2018; Häusler et al., 2018). While the detailed mechanisms of Mn carbonate formation are still debated, the Mn carbonate deposits are generally associated with intervals of prolonged reoxygenation (e.g., in the Baltic Sea, plotted on Fig. 10)(Häusler et al., 2018) and most likely benefit from microbial mediation (Henkel et al. 2019). The Baltic Sea deeps represent an extreme example of Mn shuttling, where Mo is highly enriched as it adsorbed to Mn but U and Re are much less enriched. Enrichment factors calculated relative to average
shale (Wedepohl, 1971; Wedepohl, 1991) for the Landsort Deep in the Baltic Sea is plotted on Fig. 10. Data was generated from the 36GC-4 Core (see Häusler et al., 2017; Häusler et al., 2018; Dellwig et al., 2019) for Landsort Deep core and site description and age model) and is presented in the data repository.

## 6. Conclusions and conceptual model

This study provides insight in to the sedimentation in an epicontinental seaway during a greenhouse world. Examining
sediment away from strong detrital dilution and overprint allows us to unpick the global controls on sedimentation and gives us further insight in to the key processes in this palaeogeographic setting. Far away from detrital inputs, this location provides an excellent data set for examining changes in water column processes.

The studied interval of the Kimmeridge Clay Formation in the Cleveland Basin was deposited in a highly dynamic environment that fluctuated from LVMI deposition to HVMIs (Fig. 11). During LVMI sedimentation (Fig. 11a), the water
column was oxic and was able to sustain life to a degree that the sediment was extensively bioturbated. However, the sediment pore waters were suboxic which facilitated the enrichment of TOC.

During each HVMI, the depositional environment experienced episodic and drastic changes. Trace element data indicate redox conditions of the bottom waters alternating between fully oxic and euxinic. During oxic conditions, organic matter (OM) and carbonate production proliferated (10a, b). High primary productivity led to an increase in higher trophic feeders
that produced masses of faecal pellets, in turn resulting in an increase in OM flux rate to the seafloor that aided the preservation of carbon and diluted the detrital material. At the same time, Mn and Fe (oxyhydr)oxides were deposited at the seafloor, together with adsorbed trace metals (Fig. 11b). Detrital element proxies indicate that the HVMIs, as a whole, have periodically higher depositional energies related to the LVMIs. The oxygenation events most likely related to the downward supply of oxic water by either currents or waves.

Once these higher energetic conditions ended, the supply of oxygen to the seafloor could not keep up with the oxygen demand of organic matter degradation, and the deeper waters became anoxic/euxinic. During these intervals, sedimentation

was predominated by algal macerals delivered to the sediment as organo-minerallic aggregates and marine snow under quiescent conditions. The production of sulphide minerals (e.g. pyrite) and the accumulation of related sulphide forming trace metals was enhanced under these conditions, with the Fe being resupplied by a suboxic Fe shuttle that remobilised Fe through the chemocline. At times when Fe was not supplied, OM became sulphurised, which enhanced its preservation potential. Trace element enrichment was facilitated by a Mn/Fe particulate shuttle.

We observe that the studied section bears similarity to the Landsort Deep of the present-day Baltic Sea in that a genereally suboxic to euxinic system had periodic reoxygenation events.

In the context of the published chronostratigraphic framework, alternations between LVMIs and HVMIs occur on a short eccentricity (100 kyr) timescale. This is in line with the predictions presented by Armstrong et al. (2016), whereby an orbitally modulated expansion of the Late Jurassic Hadley Cell led to alternations between humid and arid climate conditions. We propose that the highly dynamic conditions in the studied intervals are representative of wet and humid conditions during which changes in continental weathering enhanced primary productivity and led to TOC enrichment. Conversely, the LVMIs were deposited under arid conditions.

**Supplementary Data**

Supplement A: Details on sample preparation and analytical procedures.

Supplement B: Additional down-core trace metal (Cu, Zn, Cd) plots.

Supplement C: Mn/Al vs $CaCO_3$ content plot for the higher variability mudstone intervals.

Supplement D: Enrichment factors for studied section calculated relative upper continental crust (UCC: Rudnick and Gao, 2003).

**Data Availability**

Data will be made available to the public in a data repository.

**Acknowledgements**

We thank staff at the Scottish Universities Environmental Research Centre for use of the facilities and help with sample preparation. Leon Bowen is thanked for training and support on the SEM. Phil Green, Alex Charlton, and David Earley are thanked for analytical assistance and training at Newcastle University. Howard Armstrong is acknowledged for assistance in sampling and discussions in the early stages of this project. Wagner Petrographic Ltd is acknowledged for the production of thin sections. We gratefully acknowledge the International Association of Sedimentologists for a postgraduate grant award

used for the preparation of thin sections and NIGL for a grant-in-kind for the carbon isotope analyses. The work contained in this publication contains work conducted during a PhD study undertaken as part of the Natural Environment Research Council (NERC) Centre for Doctoral Training (CDT) in Oil & Gas [grant number NEM00578X/1] and is fully funded by NERC whose support is gratefully acknowledged. We thank Thomas Algeo and Jennifer McKay for constructive reviews that improved this manuscript.

# Figures

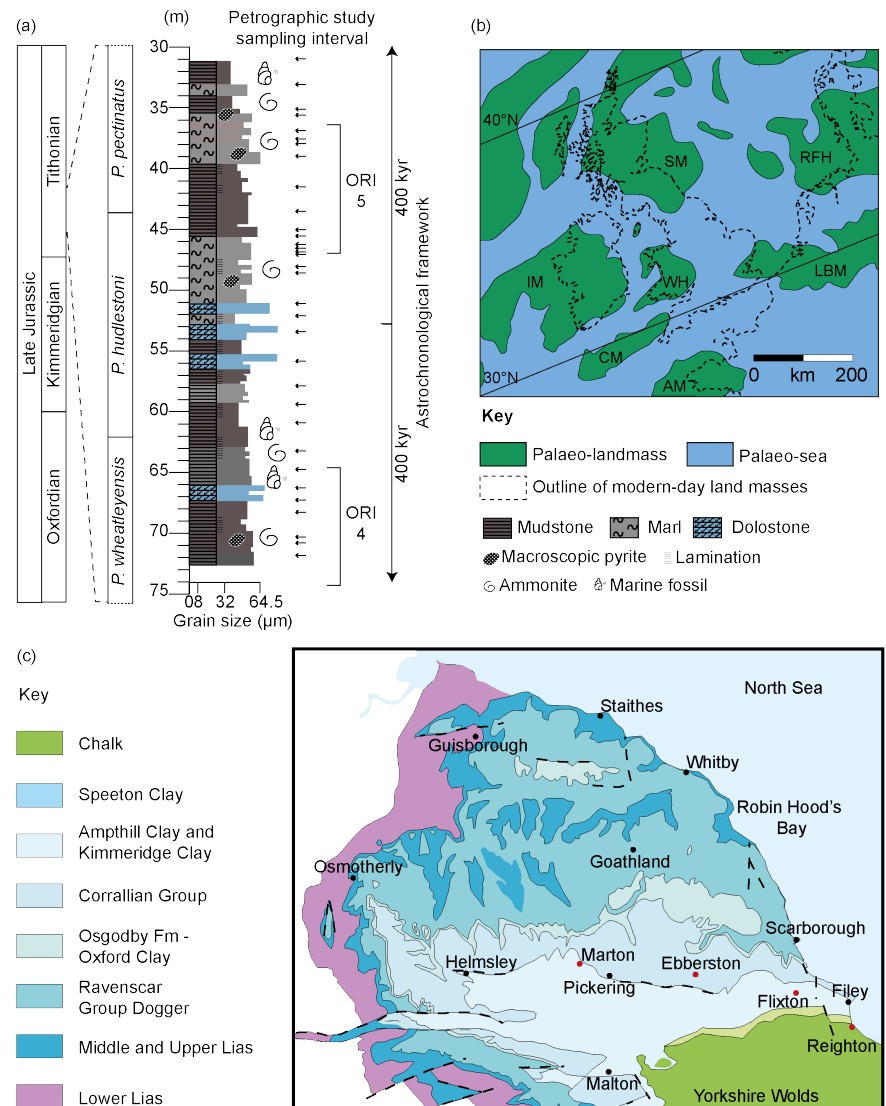

**Figure 1: (a) Graphic log showing lithogical variation of the studied section against geological time. See key for details. Horizontal arrows point to the petrographic study sample positions. Organic rich bands (ORBs) 4 and 5 as defined by Gallois (1979) Astronchronology after Huang et al., 2010 and Armstrong et al., (2016). (b) Palaeogeographical map (modified from Turner et al., 2019). (c) Modern day geological map of the Cleveland Basin (after Powell, 2010). Red dots mark the location of the Marton 87, Ebberston 87, Flixton 87, and Reighton 87 boreholes.**

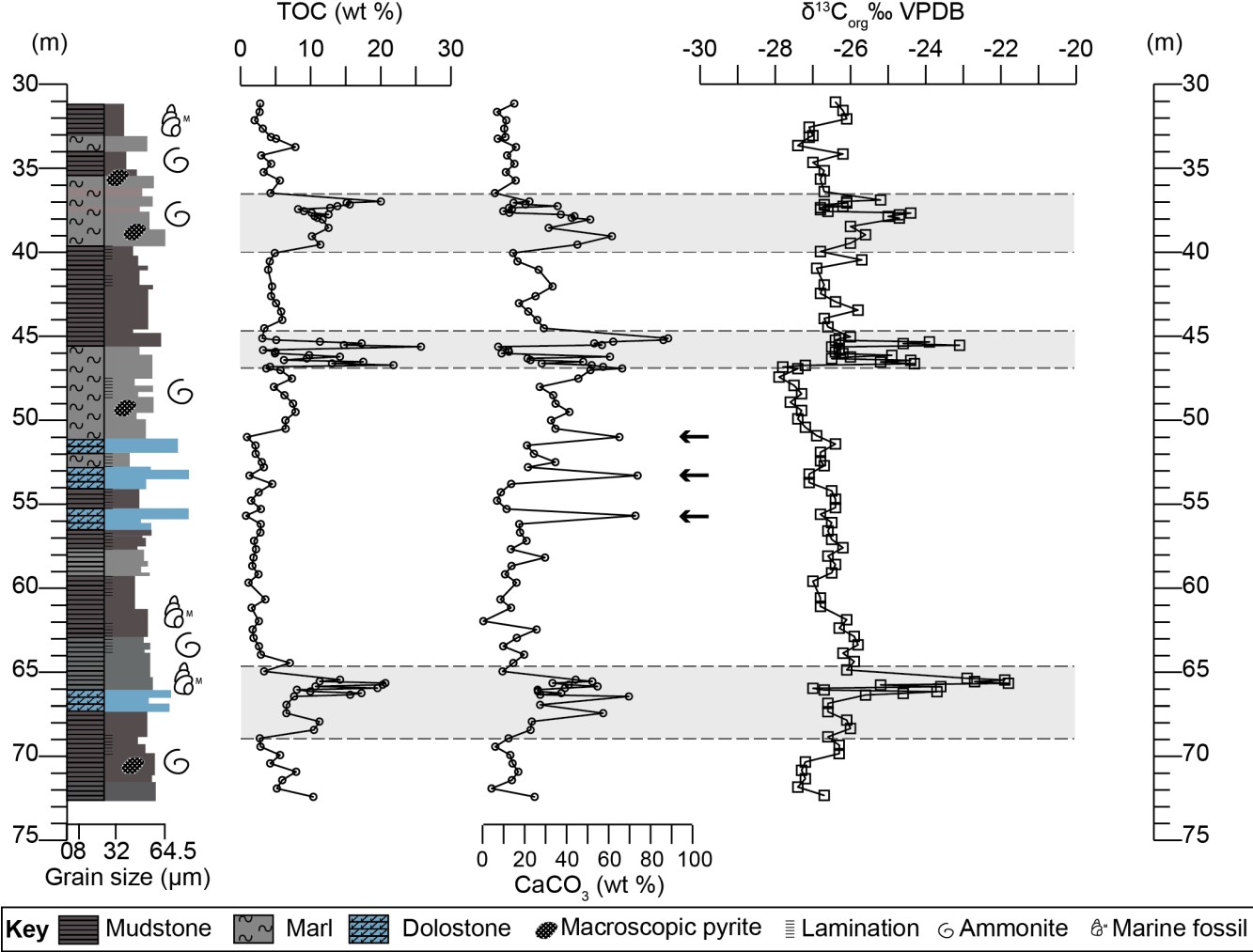

**Figure 2: Graphic log showing lithogical variation of the studied section. Depth plots of total organic carbon (TOC) contents (wt %), calcium carbonate (CaCO₃) contents (wt %), and organic carbon isotope values (‰ VPDB). Grey panels depict the higher variability intervals (HVMIs) defined in this study. Black horizontal arrows indicate samples with obvious diagenetic overprint.**

| Name | Description | Interpretation | Optical light image |
|------|-------------|----------------|---------------------|
| 1. Clastic detritus-rich medium-grained mudstone | Well churned, argillaceous matrix. Abundant medium detrital grains and calcareous nanofossils which are commonly filled with authigenic kaolinite. Equant organic material grains sit within the matrix. | Deposited in a distal setting with a constant detrital input. Bioturbation indicates well oxygenated conditions with a moderate sedimentation rate allowing for extensive faunal colonisation. Organic material is commonly type III. |  |
| 2. Organic material and calcareous pellet-rich, laminated mudstone | Discontinuous wavy lamina are organised in to normally graded beds with erosional bases . Comprised of organo-mineralic aggregates, detrital material and calcareous faecal pellets. | Organic material was deposited as algal mats that were occasionally disturbed and locally reworked. Supply of detrital material was continuous. This facies represents the highest levels of primary productivity. |  |
| 3. Coccolith-dominated medium-grained mudstone | Fine to medium normally graded calcareous mudstone beds. Dominated by coccolith plates and coccolith-rich faecal pellets in a coccolith, clay mineral and quartz rich matrix. | This facies represents times of peak carbonate productivity. Continuous supply of detrital material that was diluted by coccolith material. Material was locally reworked in to graded beds. |  |
| 4. Biogenic detritus-dominated, fine to medium-grained mudstone | Disarticulated shell fragments dominate this facies. Shells and abundant fine to coarse mud-sized quartz grains sit within an argillaceous matrix. Diagenetic calcite stringers overprint. | Deposited when depositional energy was relatively high. Shells and framework grains brought in by unusual storm activity. Calcitic stringers were formed during early diagenesis. |  |
| 5. Agglutinated foraminifera bearing, medium to coarse-grained, carbonaceous mudstone | Well churned, argillaceous and algal material matrix. Detrital grains, abundant agglutinated foraminifera, and lithic clasts sit within the matrix. | This facies represent a finely balanced system between a well mixed water productive column indicated by the foraminifera and faunal colonisation, and the production and preservation of the algal material. |  |
| 6. Carbonate cemented, coarse-grained mudstone | Medium to coarse, angular, digenetic carbonate grain-dominated sediment with an argillaceous matrix. | Dolomite rhombs formed during early diagenesis resulting from microbial sulphate reduction. These samples are discounted from paleoenvironmental interpretation. |  |

**Figure 3: Summary table describing the six lithofacies identified in the petrographic study. Key descriptions and interpretations are noted for each facies along with a representative micrograph. All images are under plain polarised light. For further description see text.**

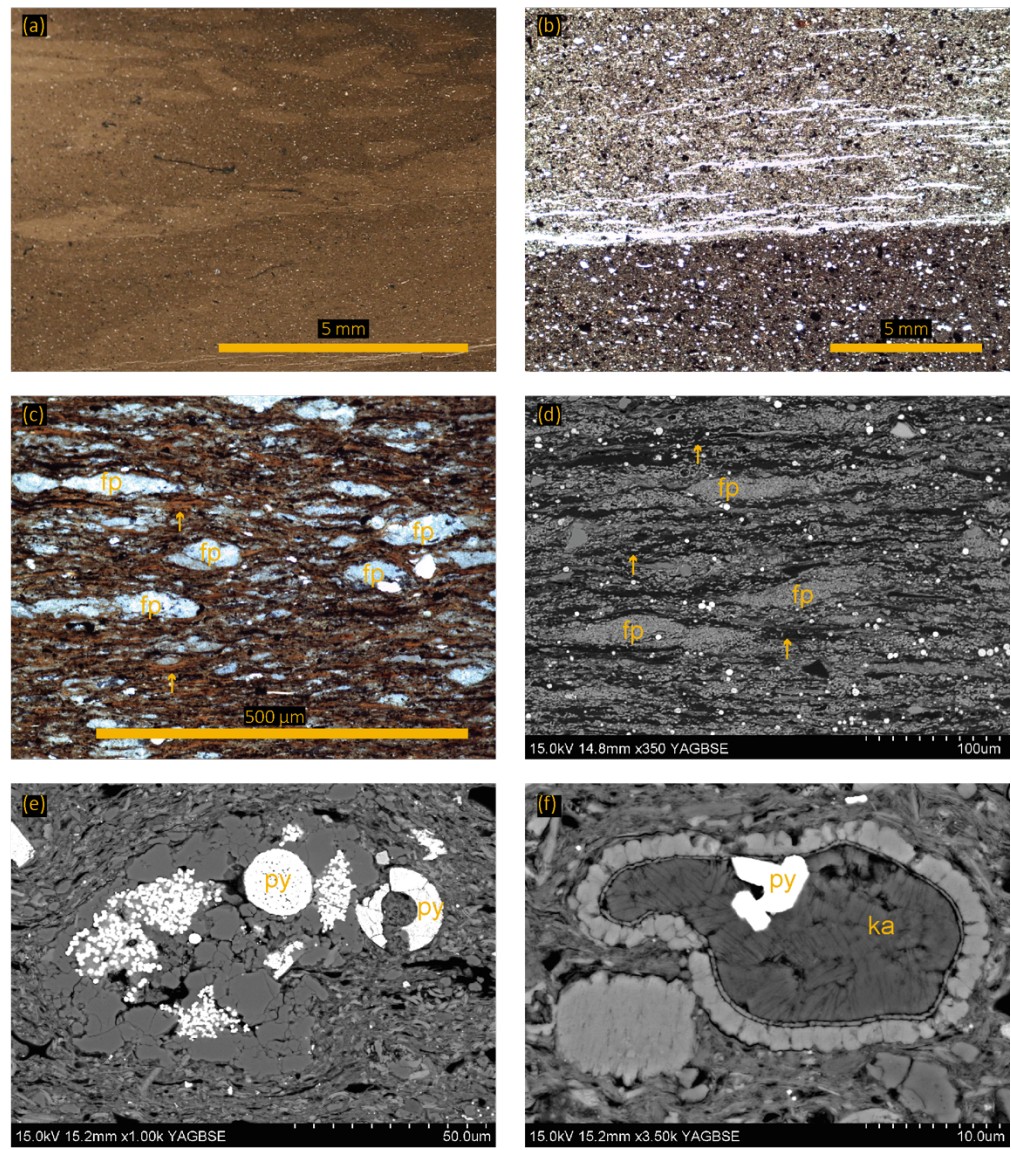

**Figure 4: a) Photomicrograph showing a bioturbated sample (lighter patches are burrows). b) Optical micrograph of an erosional base. c) Optical micrograph showing faecal pellets (fp) and algal mats (yellow arrows). d) Same sample as in (c). Scanning electron micrograph showing faecal pellets (fp) and algal mats (yellow arrows). e) Agglutinated foraminfera with pyrite (py) both as framboids and as a replacement mineral. f) Scanning electron micrograph showing a calcitic test infilled with authigenic kaolinite (ka) and pyrite (py).**

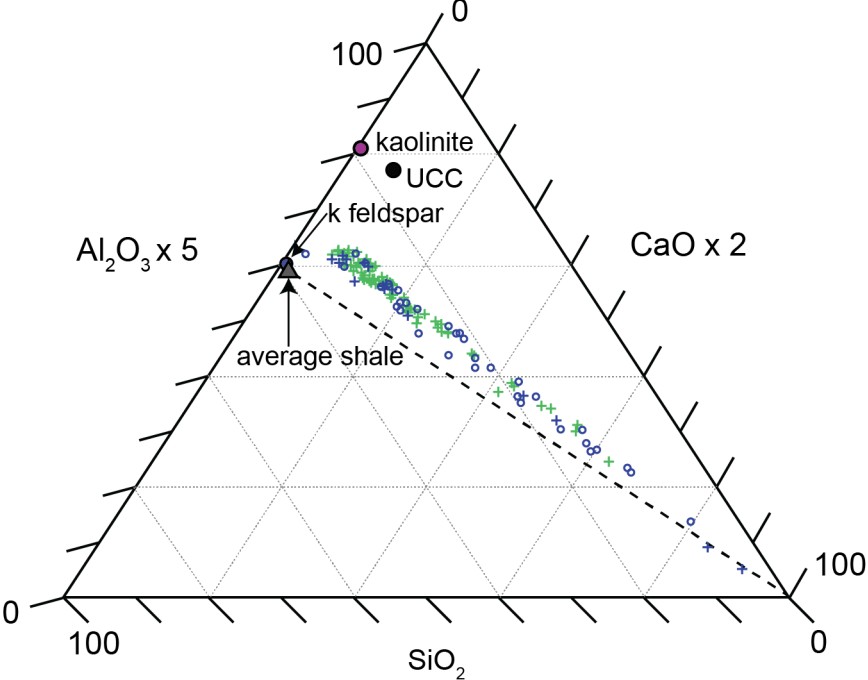

**Figure 5: Ternary diagram (Brumsack, 1989) displaying the relative contributions of clay (Al₂O₃), quartz (SiO₂), and carbonate (CaO) within all samples. Lower variability mudstone interval (LVMI) samples are indicated by green crosses and higher variability mudstone interval (HVMI) samples are marked by blue symbols, TOC-rich samples (TOC > 10 wt %) are indicated by blue circles and TOC-lean samples (TOC < 10 wt %) are indicated by blue crosses. Axes are scaled to improve display of the data. Average shale (Wedepohl, 1991), upper continental crust (UCC; Rudnick and Gao, 2003), k-feldpar and kaolinite are plotted for reference. Dashed line shows the average shale–carbonate mixing line.**

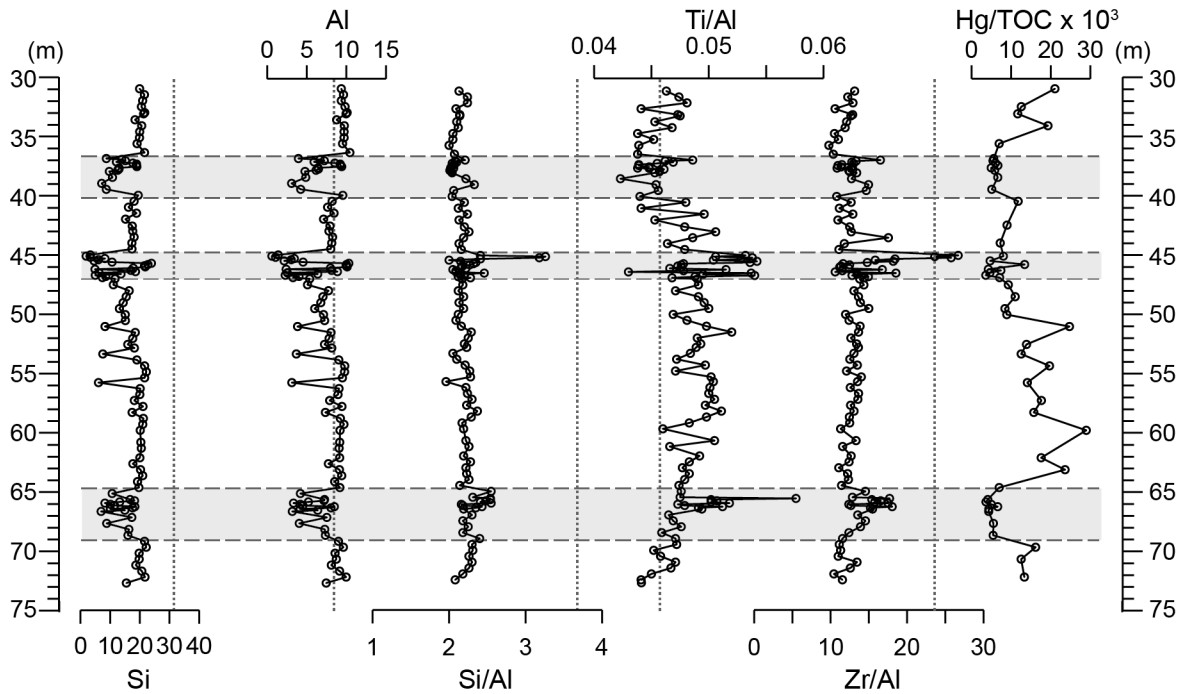

**Figure 6: Depth plots of Al and Si concentrations, element/aluminium ratios, and Hg/TOC. Si (%), Al (%), Si/Al (%/%), Ti/Al (%/%), Zr/Al (ppm/%), Hg/TOC (ppm/%). Grey panels depict the higher variability mudstone intervals (HVMIs) defined in this study. Vertical dashed lines represent upper continental crust (UCC) reference ratios.**

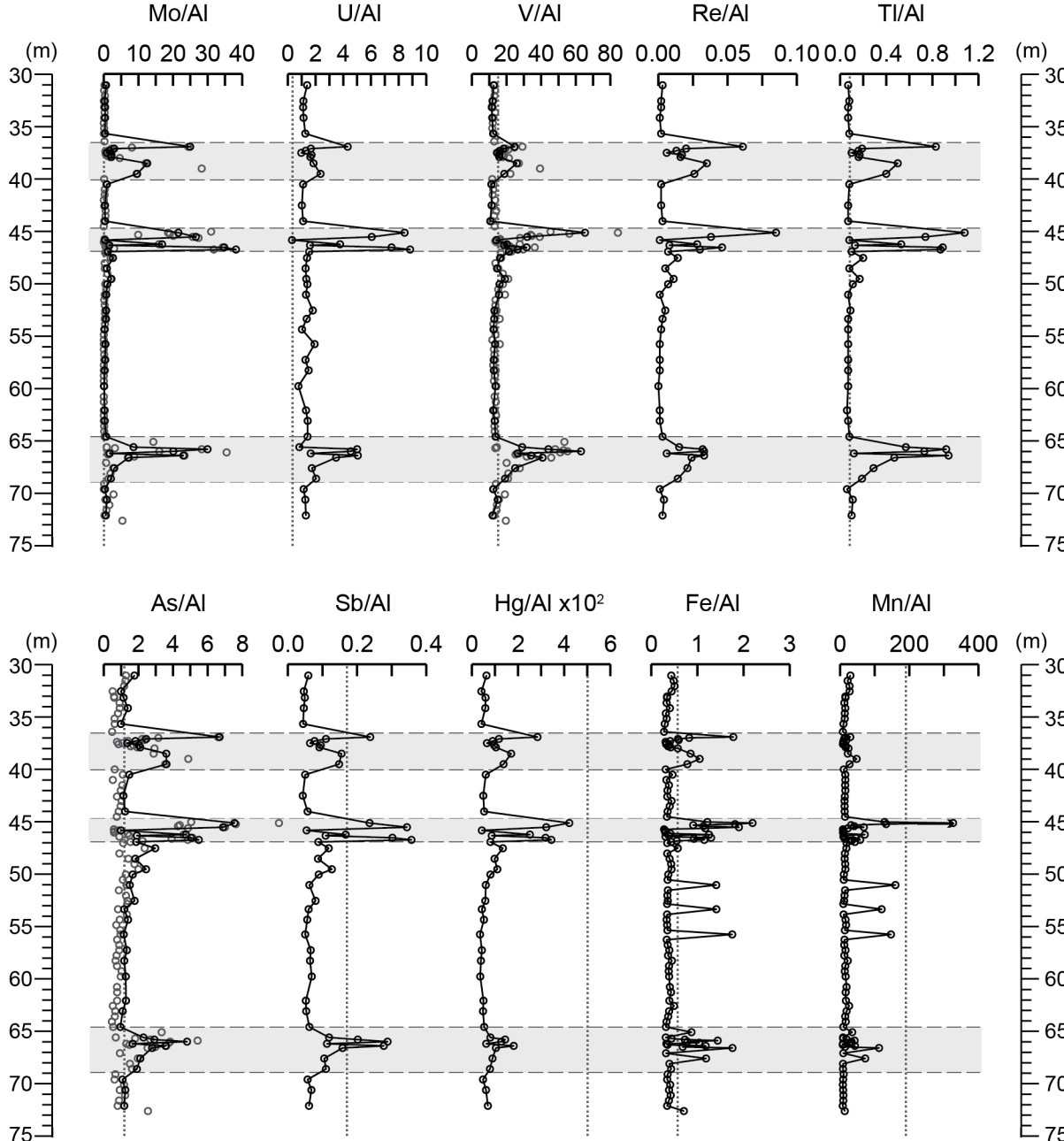

**Figure 7: Depth plots of elemental ratios used as palaeoredox indicators. Mo/Al (ppm/wt %), U/Al (ppm/wt %), V/Al (ppm/wt %), Re/Al (ppm/wt %), Tl/Al (ppm/wt %), As/Al (ppm/wt %), Sb/Al (ppm/wt %), Hg/Al x10² (ppm/wt %), Fe/Al (wt %/wt %), and Mn/Al (ppm/wt %). Grey panels depict the higher variability mudstone intervals (HVMIs) defined in this study. Where grey and black circles are seen on the same plot, grey circles mark the XRF data and black circles mark the ICP-MS data.**

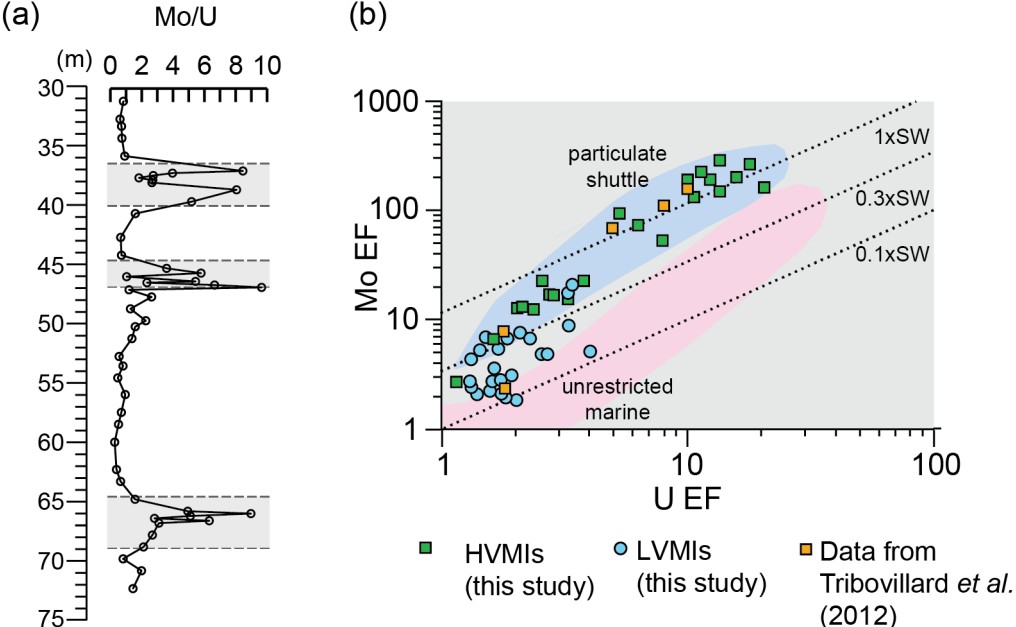

**Figure 8: a) Depth plot Mo/U (ppm/ppm) (after Algeo and Tribovillard 2009). Grey panels depict the higher variability mudstone intervals (HVMIs) defined in this study, b) scatter plot of Mo EF versus U EF calculated relative to Post Archean Average Shale (PAAS; Taylor and McLennan (1985)(after Tribovillard et al., 2012). Lower variability mudstone interval (LVMI) samples are indicated by blue circles, HVMIs are indicated by green squares. Data from the Ebberston 87 Core by Tribovillard et al., (2004) are represented by orange squares. Dashed lines are modern day seawater, 0.3 x modern day seawater and 0.1 x modern day seawater values shown for reference. The particulate shuttle is mapped in blue and unrestricted marine setting is mapped in pink. Axes are logarithmic.**

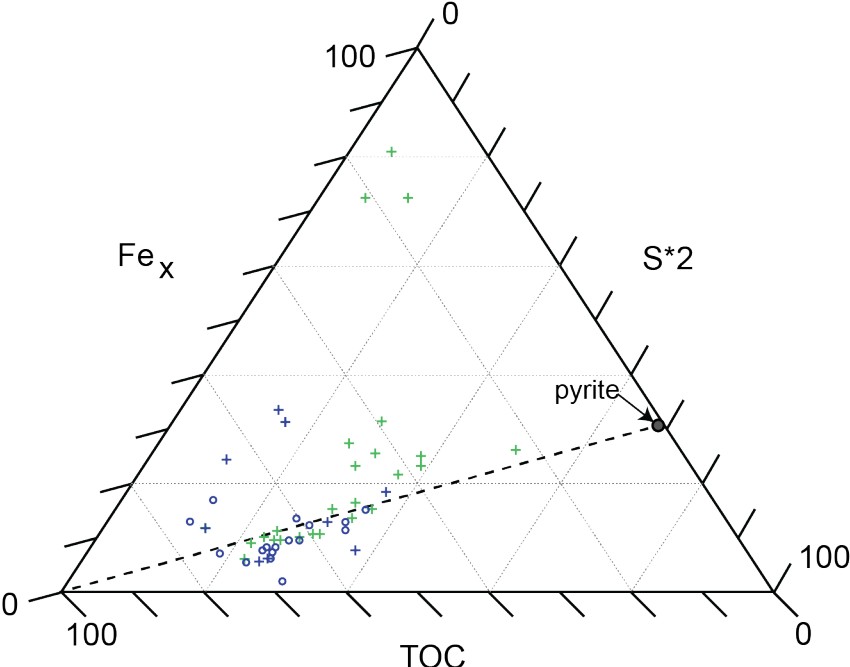

**Figure 9: Fe$_x$-TOC-S Ternary diagram (after Dellwig et al., 1999; Hetzel et al., 2011). See text for comments on Fe$_x$ calculation. Dashed line represents TOC-pyrite mixing line. Lower variability mudstone interval (LVMI) samples are indicated by green crosses (three samples nearest to the Fe$_x$ corner represent the diagenetically overprinted samples that were excluded from the interpretation). Higher variability mudstone interval (HVMI) samples are marked by blue symbols, TOC-rich samples (TOC > 10 wt %) are indicated by blue circles and TOC-lean samples (TOC < 10 wt %) are indicated by blue crosses.**

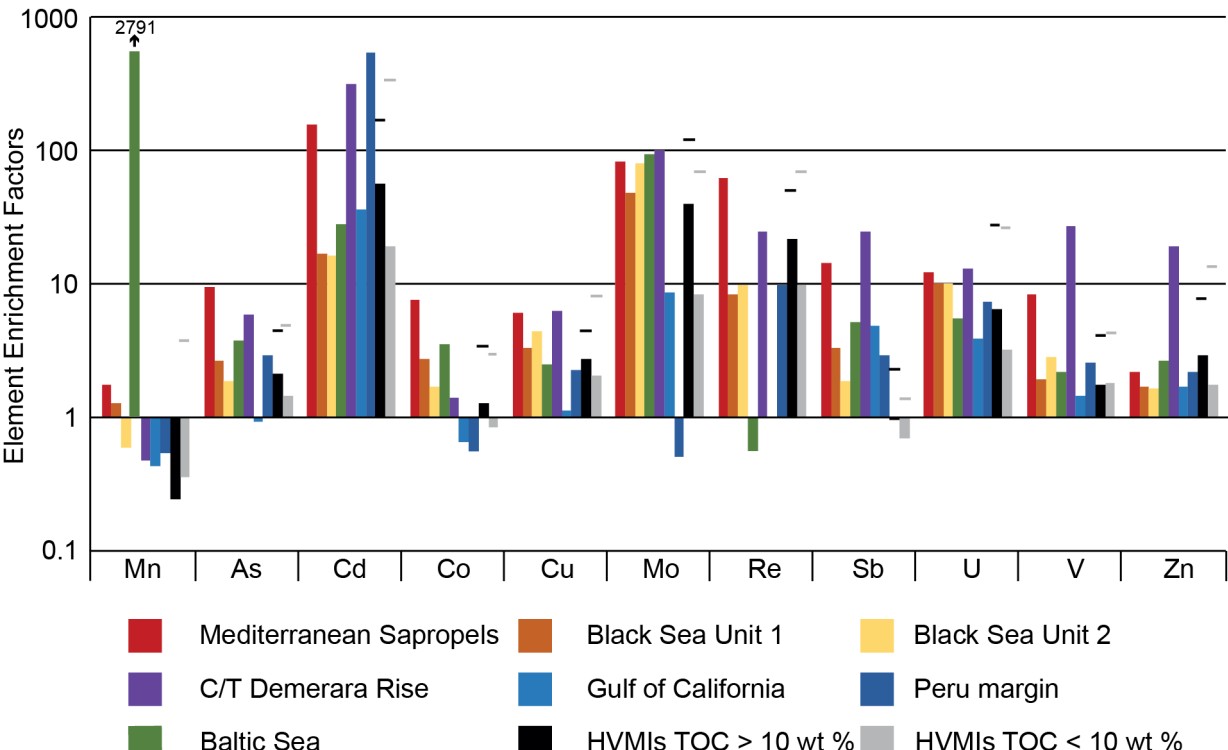

**Figure 10: Bar chart illustrating enrichment factors (EF) for Mn, As, Cd, Co, Mo, Ni, Re, Sb, U, V, and Zn relative to average shale (Wedepohl, 1991) for Mediterranean Sapropels, Black Sea Units 1 and 2, C/T Demerara Rise, Gulf of California, Peru margin (Brumsack, 2006), Baltic Sea, and higher variability mudstone intervals (HVMIs) from the present study. Maximum EFs in this study are marked by black lines.**

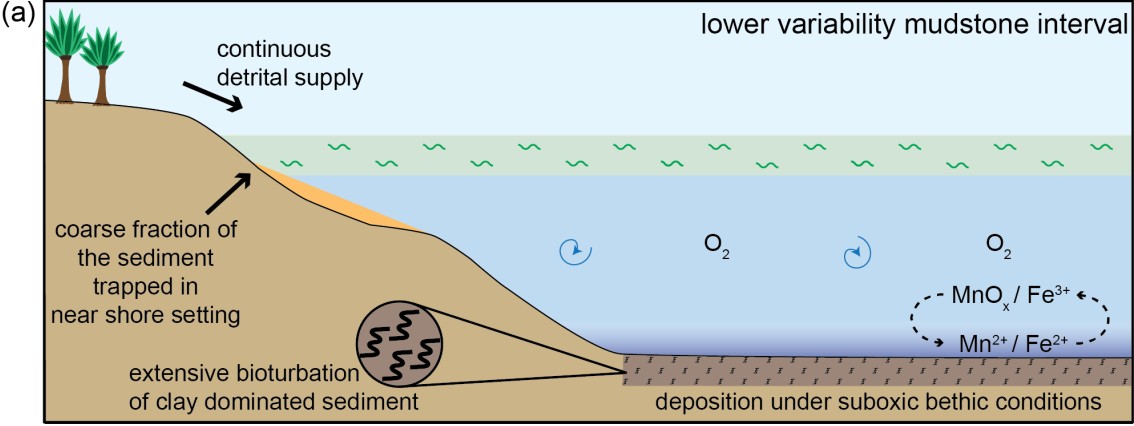

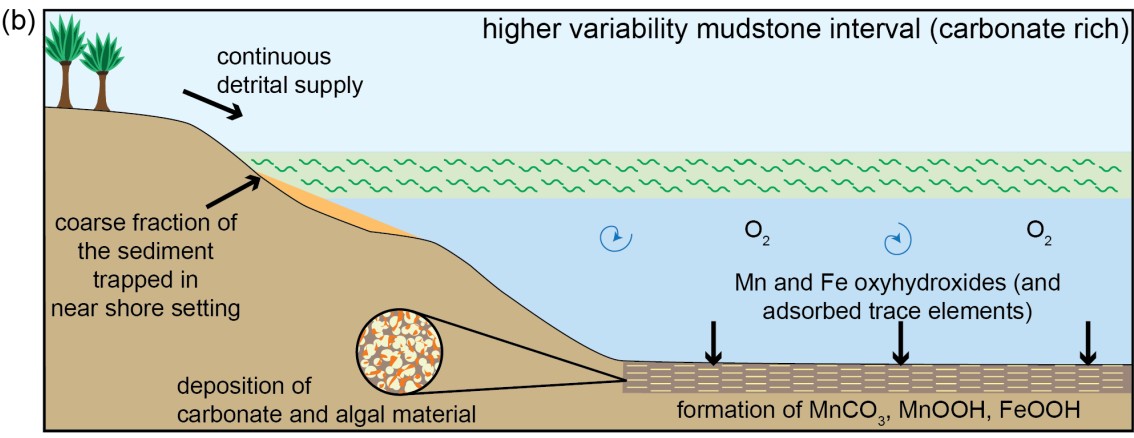

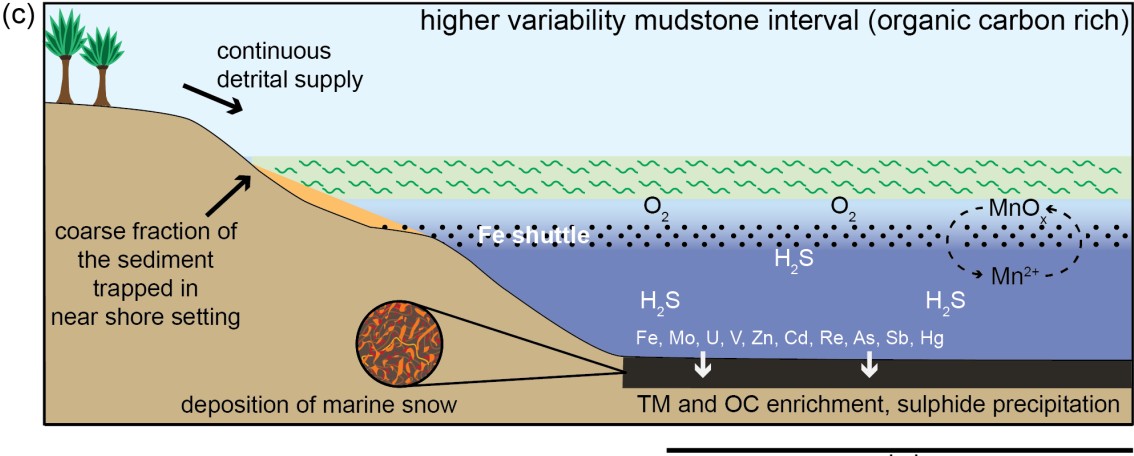

**Figure 11: Schematic diagram illustrating depositional conditions during the different intervals of sedimentation. a) Lower variability mudstone interval, b) Periods of carbonate-rich sedimentation in the higher variability mudstone intervals, c) Periods of organic carbon-rich sedimentation during the higher variability mudstone intervals. Scale bar represents bathymetric lows, generally considered to be 10–100's km across.**

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
