# Peer review of "Dynamic climate-driven controls on the deposition of the Kimmeridge Clay Formation in the Cleveland Basin, Yorkshire, UK"

_Climate of the Past, 2018_

## Author Comment (AC1) · 31 Jan 2019

We would like to note that should this manuscript be accepted we will post the datasets online in a data repository in accordance with the CoP data policy. In the meantime, we are happy to share the datasets with individuals and any person wishing to obtain an excel spreadsheet containing the data should contact Elizabeth Atar at elizabeth.atar@durham.ac.uk.

---

## Referee Comment (RC1) · Thomas Algeo (Referee) · 7 Feb 2019

This manuscript reports on a study of organic carbon (TOC), total sulfur (TS), carbonate, major and trace element contents, and carbon isotopes, together with petrographic observations, in order to investigate the primary controls on Late Jurassic sedimentation and TOC enrichment in the Cleveland Basin, Yorkshire, UK, with a view toward evaluating climatic controls on intervals of inferred high and low primary productivity.

Overall, this study is very well-executed, and the interpretations and conclusions are quite reasonable. It should become acceptable for publication following minor revision. I offer some insights on a few important issues as well as some minor comments that

the authors should consider during the revision stage.

The approach adopted in this study is to be commended in one respect in particular: It is useful to identify the main components of a sample set, and to link the geochemistry of the bulk samples to these components. This should be self-evident, but too frequently geochemical proxies are interpreted in chemostratigraphic studies with little consideration given to the underlying host sediment fractions.

Major issues:

1) Flux calculations should be presented for arguments that invoke sediment fluxes. For example: Page 8, line 31: "The fine-grained nature of the sedimentary rock may indicate a siliciclastic starvation process." When making claims regarding high or low siliciclastic or organic fluxes, it is generally a good idea to support them with some actual flux calculations. Making reliable flux calculations may be difficult for poorly dated formations, but the present study units are exceedingly well-dated, spanning the Pectinatites wheatleyensis to Pectinatites pectinatus ammonite biozones. This interval corresponds to ∼1.5 Myr (ca. 151.2-149.7 Ma) per the 2012 Geologic Time Scale (Gradstein et al., 2012, chapter 26). The study interval comprises 45 m, so its average sedimentation rate is ∼30 m/Myr–in other words, a pretty average cratonic rate. There might be condensed intervals within this succession, but it is not sediment-starved as a whole, as implied by the statement above.

Page 9, line 8: "the occurrence of normally graded beds with erosional bases in TOC-rich sections of the HVMIs indicates an energetically dynamic setting." A combination of high energy levels and sediment starvation would produce a lag deposit, i.e., concentrated high-density and/or resistant clasts such as fossil, pyrite, and/or phosphate grains. Are there any features of this type in the study succession?

Page 9, line 31: "Based on the chronostratigraphic time frame for the Yorkshire and Dorset sections (Armstrong et al., 2016; Huang et al., 2010) . . . " If there is a published astrochronology for the study formations (Huang et al., 2010), why not integrate it into

the present study and discuss its implications for the duration and accumulation rates of these formations?

2) Interpretation of controls on high TOC or high CaCO3 intervals:

Page 10, line 24: "We therefore propose that nutrient availability was the likely driver of changes in productivity. . . . The wet-dry cycles proposed by recent climate modelling (Armstrong et al., 2016) may therefore be the key driver behind oscillations in the production and preservation of TOC, i.e. the switching between the LVMIs and HVMIs." This is certainly possible but might be difficult to prove. An alternative hypothesis is that organic productivity and sinking fluxes were held more-or-less constant, and the large variations in TOC content were driven by variable influx of siliciclastics, leading to variable dilution of the organic carbon flux. Perhaps the authors could provide arguments countering this alternative hypothesis? Of course, it is also possible that both nutrient and siliciclastic fluxes were covarying in tandem.

Here is where the astrochronology of the study formations might help–if a characteristic periodic signal (e.g., 100-kyr eccentricity cycles) is present, then it is potentially possible to calculate short-term variations in sedimentation rates in a study section, rather than being limited to an average sedimentation rate for the entire section (as calculated above). An example of application of a floating time scale to analysis of short-term sedimentation rate variation is given in Algeo et al. 2011 (Algeo, T.J., Kuwahara, K., Sano, H., Bates, S., Lyons, T., Elswick, E., Hinnov, L., Ellwood, B., Moser, J. and Maynard, J.B., 2011. Spatial variation in sediment fluxes, redox conditions, and productivity in the Permian–Triassic Panthalassic Ocean. Palaeogeography, Palaeoclimatology, Palaeoecology, 308(1-2), pp. 65-83.).

Page 10, line 30: "Carbonate productivity, mainly in the form of coccoliths, varies throughout the studied KCF section and is at its maximum within the carbonate-rich sections of the HVMIs." Probably correct, but again this represents an assumption, not a proven fact, and one could argue (as for TOC variations; see above) that variable

siliciclastic influence controlled variations of carbonate content in the study section.

Page 11, line 13: "However, enrichment factors of redox sensitive trace elements (Mo and U; Fig. 8) . . . indicate that during the deposition of the LVMIs, the sediment pore water was suboxic to anoxic." This is open to interpretation, but my opinion is that the modest Mo-EFs (mean ∼4, max ∼20) and U-EFs (mean ∼2, max ∼4) of the LVMIs indicate overwhelmingly suboxic conditions. These values are not strongly supportive of anoxic conditions.

Section 5.3.4: One point about particulate shuttles that is not made clearly here is that they seem to be most effective at authigenic trace metal enrichment when redox conditions fluctuate strongly, as opposed to stably euxinic conditions (see Algeo and Tribovillard, 2009).

3) Minor issues:

Page 9, line 14: "Owing to the shallow gradients and vast extent of epicontinental seaways, sediment dispersal in the LVMIs, which are dominated by terrigenous mud, is likely to have been controlled by wind- and tide-induced bottom currents (Schieber, 2016)." Whether winds and tides could induce significant bottom currents would depend on water depths–at tens of meters, they would be important but at hundreds of meters much less so. The geologic background section (page 3, line 30) indicates considerable uncertainty regarding water depths in the NW European Sea, so the potential influence of bottom currents is uncertain.

Page 10, line 5: "Biological components (coccolithophores, foraminiferans, and organic carbon) occur in differing proportions throughout the section (Figs. 2 and 3). Our petrographic observations (Fig. 2) . . . " The presented petrographic data appear to be entirely visual/descriptive. Why not undertake point counts of organic maceral types? This would be provide more quantitative information about the nature of the organic fraction that could be compared with other data (e.g., d13C-org).

Page 10, line 22: "Water depth is not likely to have exceeded a few hundreds of meters in the distal Cleveland Basin (Bradshaw et al., 1992), suggesting that the euphotic zone could have reached the seafloor and light did not limit primary productivity." This statement betrays an incomplete understanding of the photic zone. Light intensity is attenuated quickly and drops to ∼30% of surface levels by 10 m and to a few percent by 100 m water depth in clear water; in turbid water, the rate of attenuation can be much faster with depth. Most primary productivity is typically in the upper 10 m of the water column, and there will be very little productivity at the depths suggested here.

Page 14, line 32: "we can confirm that the repeated development of anoxic/euxinic conditions in the distal Cleveland Basin was most likely due to high primary productivity, and possibly salinity stratification due to high amounts of freshwater runoff". We encourage the authors to investigate the use of paleosalinity proxies to evaluate changes in freshwater runoff in this depositional system. Check this paper for paleosalinity analysis techniques:

Wei, W., Algeo, T.J., Lu, Y., Lu, Y., Liu, H., Zhang, S., Peng, L., Zhang, J. and Chen, L., 2018. Identifying marine incursions into the Paleogene Bohai Bay Basin lake system in northeastern China. International Journal of Coal Geology, 200, pp. 1-17.

Page 15, line 14: "The HVMIs in the present study bear similarities to the Gulf of California in that they exhibit similarities in Cd enrichment and Mn depletion". The 2016 study by Tim Sweere is highly relevant in this regard and should be cited:

Sweere, T., van den Boorn, S., Dickson, A.J. and Reichart, G.J., 2016. Definition of new trace-metal proxies for the controls on organic matter enrichment in marine sediments based on Mn, Co, Mo and Cd concentrations. Chemical Geology, 441, pp. 235-245.

Page 16, line 2: "While the studied interval shares similarities and differences with both upwelling and anoxic basin type settings, we are still lacking an appropriate modern analogue. Palaeogeography exerts a fundamental control on sedimentation, in particular, TOC enrichment, but there is no modern-day example of a shallow epicontinental

seaway." Agreed, but the authors should consider the examples provided in Algeo et al. (2008):

Algeo, Thomas J., Philip H. Heckel, J. Barry Maynard, Ronald C. Blakey, Harry Rowe, B. R. Pratt, and C. Holmden. "Modern and ancient epeiric seas and the super-estuarine circulation model of marine anoxia." Dynamics of Epeiric Seas: Sedimentological, Paleontological and Geochemical Perspectives: Geological Association Canada Special Paper 48 (2008): 7-38.

Thomas Algeo University of Cincinnati State Key Labs of GPMR and BGEG, China University of Geosciences 30 January 2019

Please also note the supplement to this comment:
https://www.clim-past-discuss.net/cp-2018-172/cp-2018-172-RC1-supplement.pdf

**Supplement:**

[Figure]

**Dynamic climate-driven controls on the deposition of the Kimmeridge Clay Formation in the Cleveland Basin, Yorkshire, UK**

Elizabeth Atar[1], Christian März[2], Andrew Aplin[1], Olaf Dellwig[3], Liam Herringshaw[4], Violaine Lamoureux-Var[5], Melanie J. Leng[6], Bernhard Schnetger[7], Thomas Wagner[8].

[1] Department of Earth Sciences, Durham University, South Road, Durham, DH1 3LE, UK
[2] School of Earth and Environment, University of Leeds, Leeds, LS2 9JT, UK
[3] Leibniz-Institute for Baltic Sea Research, Marine Geology, Seestrasse 15, 18119 Rostock, Germany
[4] School of Environmental Sciences, University of Hull, Hull, HU6 7RX, UK
[5] IFP Energies Nouvelles, Geosciences Division, 1 et 4 Avenue de Bois-Préau, 92852 Rueil-Malmaison Cedex, France
[6] NERC Isotope Geosciences Laboratory, British Geological Survey, Nottingham NG12 5GG, UK and Centre for Environmental Geochemistry, School of Biosciences, University of Nottingam, Sutton Bonington Campus, Leicestershire, UK
[7] ICMB, Oldenburg University, P. O. Box 2503, 26111 Oldenburg, Germany
[8] Lyell Centre, Heriot-Watt University, Edinburgh, EH14 4AS, UK

**Abstract**

The Kimmeridge Clay Formation (KCF) is a laterally extensive, total organic carbon-rich succession deposited throughout Northwest Europe during the Kimmeridgian–Tithonian (Late Jurassic). Here we present a petrographic and geochemical dataset for a 40 metre-thick section of a well-preserved drill core recovering thermally-immature deposits of the KCF in the Cleveland Basin (Yorkshire, UK), covering an interval of approximately 800 kyr. The new data are discussed in the context of depositional processes, sediment source and supply, transport and dispersal mechanisms, water column redox conditions, and basin restriction.

[revised manuscript text omitted]
 nutrient supply from increased precipitation and continental runoff, ocean overturn, salinity/temperature stratification, and redox conditions. However, water depth reconstruction for the Laurasian Seaway are contentious and depend upon the preferred depositional model (Bradshaw et al., 1992). A water depth of tens of metres has been proposed by some authors (e.g. Hallam (1975), Aigner (1980) and Oschmann (1988)) while others (e.g. Gallois (1976), Haq et al. (1988) and Herbin et al. (1991)) have suggested a major Late Jurassic transgression that led to water depths of hundreds of metres.

[Figure]

The KCF type section is exceptionally exposed along the coast around Kimmeridge Bay, Dorset, UK (Fig. 1b) and was cored as part of a high-resolution analysis project (see e.g. Morgans-Bell et al. (2001)). It has been extensively studied by sedimentologists (Macquaker and Gawthorpe, 1993; Macquaker et al., 2010), stratigraphers (Morgans-Bell et al., 2001), palaeontologists (Lees et al., 2004), geochemists (Pearce et al., 2010), and palaeoclimatologists (Hesselbo et al., 2009) all

5    trying to unravel the processes responsible for its deposition.

Owing to a lack of coastal outcrops, however, study of laterally equivalent deposits in the Cleveland Basin (Yorkshire, UK) has been limited to four cores drilled in the 1980s (Boussafir et al., 1995; Boussafir and Lallier-Vergès, 1997; Herbin et al., 1995; Herbin and Geyssant, 1993; Herbin et al., 1991; Herbin et al., 1993; 
[revised manuscript text omitted]

In the context of developing an anoxic/euxinic water column in the Cleveland Basin with associated enrichments of redox sensitive/sulphide forming trace metals (Fig. 6), not only primary productivity but also basin restriction has to be considered as a contributing factor. A key element for the reconstruction of hydrographic basin restriction is Mo (Algeo and Lyons, 2006), as its removal from the water column and sequestration into the sediments changes dramatically under different redox conditions, more so than it is the case for other redox sensitive/sulphide forming trace metals (Helz et al., 1996).

[revised manuscript text omitted]

15   2017) for Landsorp Deep core and site description) and is presented in the data repository.

This geographical setting bears resemblance to the Late Jurassic Laurasian Seaway in that it was comprised of a series of interconnected shallow basins. Therefore, we propose that redox dynamics in the Cleveland Basin during the Late Jurassic shared similarities to those in the modern-day Baltic Sea in that they both exhibit signs of complex redox dynamics and trace metal sequestration. The enrichment factors in Fig. 9 are averaged across the HVMIs, indicating Mn depletion across these

20   zones. However, there are several samples that are enriched in Mn within the HVMIs (Fig. 6), which point towards prolonged reoxygenation events during episodes of organic enrichment in the Kimmeridge Clay Formation.

**6. Conclusions and conceptual model**

This study provides an insight in to the sedimentation in the distal parts of an epicontinental seaway during a greenhouse world. Examining sediment away from strong detrital dilution and overprint allows us to unpick the global controls on

25   sedimentation and gives us further insight in to the key processes in this palaeogeographic setting. Far away from detrital inputs, this distal location provides an excellent data set for examining changes in water column processes.

The studied interval of the Kimmeridge Clay Formation in the distal part of the Cleveland Basin was deposited in a highly dynamic environment that fluctuated from LVMI deposition to HVMIs (Fig. 10). During LVMI sedimentation (Fig. 10a), the water column was oxic and was able to sustain life to a degree that the sediment was extensively bioturbated. However, the

30   sediment pore waters were suboxic to anoxic which facilitated the enrichment of TOC.

[Figure]

During each HVMI, the depositional environment experienced episodic and drastic changes. Trace element data indicate redox conditions of the bottom waters alternating between fully oxic and euxinic. During oxic conditions, organic matter (OM) and carbonate production proliferated (10a, b). High primary productivity led to an increase in higher trophic feeders that produced masses of faecal pellets, in turn resulting in an increase in OM flux rate that aided the preservation of carbon

5 and diluted the detrital material. At the same time, Mn and Fe (oxyhydr)oxides were deposited at the seafloor, together with adsorbed trace metals (Fig. 10b). Detrital element proxies indicate that the oxygenation events were accompanied by elevated depositional energies at the seafloor, most likely related to the downward supply of oxic water by either currents or waves.

[revised manuscript text omitted]

---

## Short Comment (SC1) · 25 Mar 2019

Summary

This manuscript presents an integrated sedimentological and geochemical dataset through one section of the Kimmeridge Clay Formation (KCF) in the Cleveland Basin (Yorkshire, UK). This manuscript assesses viability of the 'Hadley cell' hypothesis, defined by intensification of wet-dry climatic cycles operating across the Boreal Seaway, as a key control on the distribution of sedimentary facies through the KCF. Six sedimentary facies (1-6) are defined. Two facies associations/packages are defined; Lower Variability Mudstone Invervals (LVMIs) and Higher Variability Mudstone Intervals

(HVMIs). Facies 1 is defined as a 'clastic' mudstone and is therefore considered the siliciclastic end-member. Facies 6 is defined as a carbonate-cemented mudstone (or possibly limestone). LVMIs almost entirely comprise Facies 1 mudstones, whereas Facies 6 is very rare. The authors suggest Facies 6 is primarily diagenetic in origin. Facies 2-5 mudstones are variously organic and pellet-rich (2), coccolith-rich (3), foraminifera-rich (4) and bioclastic (5), and typify HVMIs. LVMIs lack enrichment in redox-sensitive trace metals and are bioturbated. Whereas HVMIs typically lack bioturbation (Facies 5 is an exception) and are enriched in redox-sensitive metals. The authors utilise the distribution of redox-senstivie trace metals as palaeoredox proxies. The authors discuss and compare changing redox conditions between deposition of LVMIs and HVMIs. Related processes of sulphide production, reactive Fe input, organic matter sulphurization and chemocline stability (Fe-Mn shuttling) are considered. The authors compare anoxia in the Cleveland Basin with several modern settings. The authors conclude elevated TOC in HVMIs is best explained by coupling of bottom water anoxia to increased rates of primary productivity and increased water column stratification (freshwater cap). The model for anoxia in the Cleveland Basin is therefore a hybrid of productivity-driven and restriction-driven settings.

General Comments

The dataset underlying this manuscript is high-resolution (n=116, spanning 40 m core) and diverse (47 thin sections, TOC measurements, major and trace element geochemistry via XRF and ICM-MS and $\delta$13Corg). The underlying scientific principles and assumptions are also robust. Aspects of the manuscript itself are robust and do not require significant modification. The comparison with modern settings is useful too. Therefore there are many merits of this research. If seen through to publication, I believe this is a valuable contribution that will be of interest to a wide readership. Despite this, there are several aspects of this manuscript which require significant revision prior to publication. On this basis I recommend acceptance of this manuscript with major revisions.

My main concerns are provided in the section below. I have provided as much detail as possible, in order to help the authors improve the manuscript. I will respond to requests for clarification/discussion on any of these points if needed.

Main Concerns

- This manuscript poorly integrates sedimentological (petrographic) observations with the geochemistry. The authors define 6 sedimentary facies, yet it is very difficult to relate these facies to the geochemistry. This is primarily because the sedimentary logs are defined using 'mudstone', 'marl' and 'limestone' yet this nomenclature is not used in the main text. Therefore this is a problem of consistency between the main text and the log (Fig. 2). The sedimentary log should be defined to the facies scale. Doing so will better link the sedimentological observations with the geochemistry. Doing this may also help delineate more subtle relationships between the geochemistry and particularly Facies 2-5 within HVMIs, which might help support/develop the conclusions of this research. In addition, a bioturbation index (Lazar et al. 2015, as is referenced in this manuscript) should be plotted alongside the logs. In my opinion, for a manuscript that is focussed on bottom water redox conditions, this is very important. I also strongly recommend plotting TOC, carbonate content and $\delta$13Corg alongside some of the other redox-sensitive elements.

- Sedimentology – 1) In my opinion the facies descriptions need much more support. There are too many assertions and not enough descriptions supported by data. I think this manuscript would greatly benefit from more example microphotographs. Perhaps all that is required is some rewording to improve clarity and more detail (including logic for facies definition and ordering) plus one additional figure, which provides evidence for the following; bioturbation, phosphate clasts, faecal pellets, algal mats, pyrite microtextures, depositional processes (normal grading, erosive bases) and authigenic clays. Perhaps this could include scanning electron microphotographs too. (SEM is mentioned in the methods.) 2) Quantification – it would be advantageous if the authors quantify the abundance of all sedimentary components (e.g., carbonate, bioclasts, coccoliths, etc.) where possible – even if this is simply done at percentile resolution. (i.e., trace, 25%, 50%, 75% and <> where needed). In some cases it would be useful to quantify grain/cryst diameters (e.g., framboids), even if such observations are approximations;

- Organic matter typing – In my opinion much more emphasis should be given to the $\delta$13Corg data as a proxy for bulk OM type. I am not convinced petrographic observations are sufficiently robust/reliable in order to define bulk OM type (i.e., Type II vs III). In my opinion there are too many assertions rather than observations supported by data (in exactly the same way as the general sedimentological descriptions). Assessment of bulk OM using petrography would need to be supported by multiple annotated microphotographs for each facies. Thus in my opinion, the $\delta$13Corg record should be emphasised and utilised as the primary proxy for OM type. Petrographic observations (at the level of detail presented) are secondary;

- Algal mats – this is very interesting but needs to be supported by evidence – ideally microphotographs and then integrated onto Fig. 10;

- Evidence for sediment starvation & winnowing (e.g., p9, line 10) – this needs to be supported by evidence (i.e., microphotographs). Perhaps it is also appropriate to estimate a mean sediment accumulation rate for this section too. i.e., the fine-grained nature of this section does not necessarily indicate this was a distal setting subject to low sediment accumulation rates;

- Enrichment factors – PAAS or UCC? It is unclear whether the authors calculated enrichment factors using PAAS or UCC, which are not the same. The manuscript body quotes UCC but some of the figure captions quote PAAS. In my opinion, it is preferable to calculate enrichment factors using PAAS, as this allows for comparison with many other black shale studies. PAAS vs. UCC is likely to be particularly important at low EFs (close to 1) when plotted onto the log-log Mo and U EF cross-plot of Tribovillard et al. (2012) Chemical Geology (Fig. 8b in this manuscript). I recommend re calculation

of EFs using PAAS if necessary.

- This manuscript lacks detailed comment on sea level. Was sea level stable or fluctuating? If fluctuating, was this via eustacy or a local mechanism? Sea level (but not fluctuation through the section) is mentioned briefly in the geological setting and also briefly mentioned in the discussion - p14 line 23-24 "...concluded that TOC enrichment in the Cleveland Basin occurred during times of transgression.." Yet the authors do not provide the context, or critique, of sea level fluctuation as a potential control on the distribution of sedimentary facies and organic matter through the KCF. It is extremely important the authors give more consideration to the role of sea level fluctuation. This should feature in the geological setting and discussion, and perhaps also in the introduction. Could HVMIs represent transgressive and highstand packages? Therefore couldn't the 'cyclicity' between HVMIs and LVMIs simply relate to sea level fluctuation? i.e., an 'expanding puddle' model of anoxia (e.g., Wignall, 1994, Black Shales). If sea level fluctuation is not considered the key control on the distribution of facies, the authors should explain why this is the case.

- The model for anoxia. (1) In my opinion, I think there is a more subtle signal through each LVMI and into the overlying HVMI. This is particularly clear, in my opinion, when assessing the $\delta13Corg$ record (Fig. 2). Can the authors consider my suggestion? My argument is this. Increasingly wet conditions through deposition of each LVMI drives increasing run-off, input of TOM and progradation of a freshwater cap. Ultimately basin stratification approaches a tipping point, which triggers bottom water anoxia, during the extreme wet part of the cycle. Bottom water anoxia drives the 'eutrophication pump' (e.g., Sageman et al., 2003, Chemical Geology), generating a positive feedback in terms of productivity and further expansion of anoxic conditions in bottom waters. The euphotic zone is no longer P-limited. Perhaps somehow these conditions encouraged carbonate productivity oscillation. Then finally, progressive reduction in precipitation as the onset of the 'dry' part of the cycle reduces freshwater input, therefore gradually weakening the pycnocline. Ultimately this process encourages ventilation of bottom

waters. Bottom water ventilation switches off the 'eutrophication pump', reducing productivity, further reducing the OM load to seabed and further promoting ventilation. Then the cycle starts again. At the least, I think the role of the 'eutrophication pump' deserves comment. (2) Bottom water conditions were apparently 'intermittently euxinic' during deposition of organic-rich parts of HVMIs rather than 'permanently euxinic'. The data presented indicates an unstable chemocline (particulate shuttle) and in my opinion does not indicate a strongly stratified and permanently euxinic system. Therefore Fig. 10 and relevant discussion should be revised to reflect this.

- The prose should be improved throughout this manuscript. For example, many sentences are too long and/or poorly structured. I have highlighted some specific cases in the annotated PDF. In many cases this can be resolved quickly, by splitting one long sentence into 2 or 3 shorter sentences. This will help communicate the science. In places the language is too informal and mixes tenses.

Please also see the attached PDF for detailed annotations – the figures, in particular, could be improved substantially.

Joe Emmings, British Geological Survey, 25/03/19

Please also note the supplement to this comment:
https://www.clim-past-discuss.net/cp-2018-172/cp-2018-172-SC1-supplement.pdf
* * *
[Figure]

**Supplement:**

[Figure]

[Figure]

**Dynamic climate-driven controls on the deposition of the Kimmeridge Clay Formation in the Cleveland Basin, Yorkshire, UK**

Elizabeth Atar[1], Christian März[2], Andrew Aplin[1], Olaf Dellwig[3], Liam Herringshaw[4], Violaine Lamoureux-Var[5], Melanie J. Leng[6], Bernhard Schnetger[7], Thomas Wagner[8].

5   [1] Department of Earth Sciences, Durham University, South Road, Durham, DH1 3LE, UK
[2] School of Earth and Environment, University of Leeds, Leeds, LS2 9JT, UK
[3] Leibniz-Institute for Baltic Sea Research, Marine Geology, Seestrasse 15, 18119 Rostock, Germany
[4] School of Environmental Sciences, University of Hull, Hull, HU6 7RX, UK
[5] IFP Energies Nouvelles, Geosciences Division, 1 et 4 Avenue de Bois-Préau, 92852 Rueil-Malmaison Cedex, France
10   [6] NERC Isotope Geosciences Laboratory, British Geological Survey, Nottingham NG12 5GG, UK and Centre for Environmental Geochemistry, School of Biosciences, University of Nottingam, Sutton Bonington Campus, Leicestershire, UK
[7] ICMB, Oldenburg University, P. O. Box 2503, 26111 Oldenburg, Germany
[8] Lyell Centre, Heriot-Watt University, Edinburgh, EH14 4AS, UK

15   **Abstract**

The Kimmeridge Clay Formation (KCF) is a laterally extensive, total organic carbon-rich succession deposited throughout Northwest Europe during the Kimmeridgian–Tithonian (Late Jurassic). Here we present a petrographic and geochemical dataset for a 40 metre-thick section of a well-preserved drill core recovering thermally-immature deposits of the KCF in the Cleveland Basin (Yorkshire, UK), covering an interval of approximately 800 The new data are discussed in the context
20   of depositional processes, sediment source and supply, transport and dispersal mechanisms, water column redox conditions, and basin restriction.

[revised manuscript text omitted]

In the context of developing an anoxic/euxinic water column in the Cleveland Basin with associated enrichments of redox
5   sensitive/sulphide forming trace metals (Fig. 6), not only primary productivity but also basin restriction has to be considered
   ⓒ contributing factor. A key element for the reconstruction of hydrographic basin restriction is Mo (Algeo and Lyons, 2006), as its removal from the water column and sequestration into the sediments changes dramatically under different redox conditions, more so than it is the case for other redox sensitive/sulphide forming trace metals (Helz et al., 1996).

[revised manuscript text omitted]

15 2017) for Landsorp Deep core and site description) and is presented in the data repository.

This geographical setting bears resemblance to the Late Jurassic Laurasian Seaway in that it was comprised of a series of interconnected shallow basins. Therefore, we propose that redox dynamics in the Cleveland Basin during the Late Jurassic shared similarities to those in the modern-day Baltic Sea in that they both exhibit signs of complex redox dynamics and trace metal sequestration. The enrichment factors in Fig. 9 are averaged across the HVMIs, indicating Mn depletion across these

20 zones. However, there are several samples that are enriched in Mn within the HVMIs (Fig. 6), which point towards prolonged reoxygenation events during episodes of organic enrichment in the Kimmeridge Clay Formation.

**6. Conclusions and conceptual model**

This study provides an insight in to the sedimentation in the distal parts of an epicontinental seaway during a greenhouse world. Examining sediment away from strong detrital dilution and overprint allows us to unpick the global controls on

[revised manuscript text omitted]

---

## Author Comment (AC3) · 5 May 2019

We thank Joe Emmings for taking the time to provide detailed comments and alternative interpretations of the dataset. In order to keep this response clear and concise, we have grouped comments from SC1 and the supplementary material into 7 categories. Where comments were repeated or of a similar nature, we address the first comment (but also refer to the latter ones). Comments from SC1 are in blue, comments from the SC1 supplementary information are in purple, our responses are in black text.

**1.1 Integration of geochemistry and sedimentology**

This manuscript poorly integrates sedimentological (petrographic) observations with the geochemistry. The authors define 6 sedimentary facies, yet it is very difficult to relate these facies to the geochemistry. This is primarily because the sedimentary logs are defined using 'mudstone', 'marl' and 'limestone' yet this nomenclature is not used in the main text. Therefore this is a problem of consistency between the main text and the log (Fig. 2). The sedimentary log should be defined to the facies scale. Doing so will better link the sedimentological observations with the geochemistry. Doing this may also help delineate more subtle relationships between the geochemistry and particularly Facies 2-5 within HVMIs, which might help support/develop the conclusions of this research.

The sedimentary log is redrawn from the initial work on the core (ca. 1987; documented in IFPEN internal files), which was carried out using hand specimens. The facies model presented in this study was constructed using both hand specimens and thin sections of selected intervals only, so it is impossible to construct a sedimentary log from the latter technique that is directly comparable to the original 1987 log. We wanted to incorporate all the work that was done on this core in our manuscript, and in doing so consider it appropriate to use both the original and our new datasets, even if they are not directly comparable/correlative. To aid interpretation, we will add an appendix that details the assignment of individual thin sections to their respective facies.

Petrographic observations (at the level of detail presented) are secondary
This manuscript is mainly a geochemical study that is supported by petrographic observations, so we regard the balance between both types of observations/data as appropriate.

**1.2 Sedimentology**

- In addition, a bioturbation index (Lazar et al. 2015, as is referenced in this manuscript) should be plotted alongside the logs. In my opinion, for a manuscript that is focussed on bottom water redox conditions, this is very important.
- Please add bioturbation record to the logs (Figs. 2, 5, 6) - this is very important (page 6, line 9)
- Please add bioturbation indices (see Lazar et al. 2015) on the logs (page 7, line 4)
We agree that this will aid interpretation and have added it to Fig. 2.

I also strongly recommend plotting TOC, carbonate content and d13Corg alongside some of the other redox-sensitive elements.
While we agree that this could aid interpretation, we want to avoid repetition of data plots in the manuscript to not make already busy figures even busier; we therefore added grey horizontal panels to all depth plots to facilitate comparisons/correlation between figures.

- In my opinion the facies descriptions need much more support. There are too many assertions and not enough descriptions supported by data. I think this manuscript would greatly benefit from more example microphotographs. Perhaps all that is required is some rewording to improve clarity and more detail (including logic for facies definition and ordering) plus one additional figure, which provides evidence for the following; bioturbation, phosphate clasts, faecal pellets, algal mats, pyrite microtextures, depositional processes (normal grading, erosive bases) and authigenic clays. Perhaps this could include scanning electron microphotographs too. (SEM is mentioned in the methods.)
- Algal mats – this is very interesting but needs to be supported by evidence – ideally microphotographs and then integrated onto Fig. 10
- Personally I think this section would benefit from at least one further figure of example microphotographs supporting the features described here. The examples in Fig. 3 are good but in my opinion more support is required. (page 5, line 21)

We agree and have added another figure to the manuscript comprising optical micrographs and SEM images to illustrate the features outlined above.

Evidence for sediment starvation & winnowing (e.g. page 9, line 10) – this needs to be supported by evidence (i.e., microphotographs).

We agree that this issue was not ideally phrased, which was also highlighted by Reviewer 1's comments on the same matter, and we have updated our interpretation in the revised manuscript.

Perhaps it is also appropriate to estimate a mean sediment accumulation rate for this section too. i.e., the fine-grained nature of this section does not necessarily indicate this was a distal setting subject to low sediment accumulation rates

We agree. Following Reviewer 1's comments/discussion of calculated sedimentation rates, we have incorporated them into the revised manuscript.

- Quantification – it would be advantageous if the authors quantify the abundance of all sedimentary components (e.g., carbonate, bioclasts, coccoliths, etc.) where possible – even if this is simply done at percentile resolution. (i.e., trace, 25%, 50%, 75% and <> where needed). In some cases it would be useful to quantify grain/cryst diameters (e.g., framboids), even if such observations are approximations.
- Can the authors quantify proportions? What does 'dominated' mean in terms of %? Doing this will add much more value to this manuscript, and will help readers compare this work with their own

Adding approximations of sedimentary components counted in thin sections increases the errors on the dataset. Instead, we have used geochemistry, supported by petrographic observations, to quantify sedimentary components. For example, $CaCO_3$ (measured by LECO) gives a quantitative measurement of carbonate within each sample. However, we have added details on the sizes of individual sedimentary components (e.g. framboids) where we had not done so in the original manuscript.

Organic matter typing – In my opinion much more emphasis should be given to the d13Corg data as a proxy for bulk OM type. I am not convinced petrographic observations are sufficiently robust/reliable in order to define bulk OM type (i.e., Type II vs III). In my opinion there are too many assertions rather than observations supported by data (in exactly the same way as the

general sedimentological descriptions). Assssment of bulk OM using petrography would need to be supported by multiple annotated microphotographs for each facies. Thus in my opinion, the d13Corg record should be emphasised and utilised as the primary proxy for OM type.

We agree that geochemical data is required to support petrographic observations determining OM type. We have updated the manuscript to more strongly reflect the relationship between OM and $\delta^{13}C_{org}$ and incorporated published RockEval pyrolysis data to further support our interpretations.

Can the authors comment on the nature of the contacts between LVMIs and HVMIs? For example are the contacts sharp or gradational?

Given the sampling strategy employed (i.e. sampling representative intervals of macroscopically distinguishable lithologies) and the often subtle nature of facies changes, it is not possible to comment on the nature of the boundaries between facies.

Also what is the difference between a mudstone that is 'organic material... rich' (Facies 2) and 'carbonaceous' (Facies 4)

The difference lies in other aspects of the facies, in that Facies 2 contains calcareous pellets and Facies 4 contains agglutinated foraminifers.

(with reference to authigenic kaolinite) Interesting. Is this booky? (page 6, line 7)

Yes, we now include photomicrograph illustrating this.

**1.3 Geochemistry**

Enrichment factors – PAAS or UCC? It is unclear whether the authors calculated enrichment factors using PAAS or UCC, which are not the same. The manuscript body quotes UCC but some of the figure captions quote PAAS. In my opinion, it is preferable to calculate enrichment factors using PAAS, as this allows for comparison with many other black shale studies. PAAS vs. UCC is likely to be particularly important at low EFs (close to 1) when plotted onto the log-log Mo and U EF cross-plot of Tribovillard et al. (2012) Chemical Geology (Fig. 8b in this manuscript). I recommend re calculation.

We are aware of the differences between PAAS and UCC. When looking at downcore trends, we normalise our element contents to UCC because it most closely matches the lithogenic background sedimentation at our coring site. PAAS is only used in Fig 8b (Mo-U enrichment factor cross plot), as this facilitates direct comparison to the original data generated by Tribovillard (1994) using PAAS for normalisation. For the comparison to other organic-rich deposits (Fig. 9), we normalise to Average Shale (AS; Wedepohl, 1991) following Brumsack (2006) who published the most comprehensive data base on such deposits. Thus, normalising to UCC, PAAS, and AS each has its own merits depending on the purpose of normalisation. We have updated the manuscript to make this clearer to the reader.

- The model for anoxia. (1) In my opinion, I think there is a more subtle signal through each LVMI and into the overlying HVMI. This is particularly clear, in my opinion, when assessing the 13Corg record (Fig. 2). Can the authors consider my suggestion? My argument is this. Increasingly wet conditions through deposition of each LVMI drives increasing run-off, input of TOM and progradation of a freshwater cap. Ultimately basin stratification approaches a tipping point, which triggers bottom water anoxia, during the extreme wet part of the cycle. Bottom

water anoxia drives the 'eutrophication pump' (e.g., Sageman et al., 2003, Chemical Geology), generating a positive feedback in terms of productivity and further expansion of anoxic conditions in bottom waters. The euphotic zone is no longer P-limited. Perhaps somehow these conditions encouraged carbonate productivity oscillation. Then finally, progressive reduction in precipitation as the onset of the 'dry' part of the cycle reduces freshwater input, therefore gradually weakening the pycnocline. Ultimately this process encourages ventilation of bottom waters. Bottom water ventilation switches off the 'eutrophication pump', reducing productivity, further reducing the OM load to seabed and further promoting ventilation. Then the cycle starts again. At the least, I think the role of the 'eutrophication pump' deserves comment. (this comment is slightly rephrased on page 10, line 25)

The $\delta^{13}C_{org}$ record through the LVMIs reflects the global carbon isotope curve mapped at multiple sites across the world (Gröcke et al., 2003). We have updated the manuscript to include this. The eutrophication pump was most likely active during formation of the organic-rich intervals (it mostly is during times of bottom water anoxia in shallow water masses), but we do not have the data to judge whether it was the dominant factor controlling organic matter accumulation and/or redox conditions. This would require additional information about salinity stratification, phosphorus speciation, nitrogen concentrations/isotopes.

Bottom water conditions were apparently 'intermittently euxinic' during deposition of organic-rich parts of HVMIs rather than 'permanently euxinic'. The data presented indicates an unstable chemocline (particulate shuttle) and in my opinion does not indicate a strongly stratified and permanently euxinic system. Therefore Fig. 10 and relevant discussion should be revised to reflect this.

We have implemented this recommendation in the revised manuscript.

Neither excess Fe or early diagenetic pyrite oxidation preclude sulphurization. We observe exactly the reverse in the Bowland Shale. Paradoxically, sulphurization is triggered by Fe loading in some environments. We have a ms accepted for publication on this topic, available on request, if the authors would like to see this. (I do not think the Bowland Shale model for anoxia is analogous to the KCF, but nonetheless, present research shows sulphurization does not require Fe-limitation and perhaps this can be acknowledged.)

We have updated the manuscript to include recent references on early sulphurisation of OM despite the presence of reactive Fe. While the relationship between reactive Fe availability and the sulphurisation of OM can be more complex than previously assumed, detailed investigation of this matter is beyond the scope of this manuscript.

The authors could assess the viability of normalising to Al, by calculating the coefficient of variation - see Tribovillard et al. (2006)

Normalising to Al is a valid and common approach, and only problematic at very low Al contents. We do not use element/Al ratios in a quantitative sense, as we are looking at patterns and how they change, and so our approach is sufficient for this purpose.

The authors imply Mn enrichment indicates oxygenation - but this is not necessarily the case - Mn can be fixed in sulphide under sulphidic conditions. See Lyons and Severmann 2006 Geochimica et. Cosm. Acta., for example (page 12, line 3).

MnS is extremely rare and in the modern environment is only seen in some very particular deep basins of the Baltic Sea – but even there, the vast majority of Mn is bound as carbonates that

formed following oxygenation of a Mn²⁺ charged water column, and later burial and reduction of the Mn oxides.

Can the authors comment on why KCF Mn is so low compared to the other modern examples? (page 27)

Either there wasn't a lot of Mn²⁺ accumulated in the water column (since the basin was not completely restricted and Mn oxides might have precipitated in oxic shallower environments), or some of the precipitated Mn oxides were reduced fairly close to the sediment-water interface and could diffuse back into the water column, or the oxidation events occurred between fairly short periods of anoxia/euxinia and only little Mn²⁺ was available in the water column for oxidation. Our data do not support any one of these arguments over the others.

**1.4   Sea level**

- This manuscript lacks detailed comment on sea level. Was sea level stable or fluctuating? If fluctuating, was this via eustacy or a local mechanism? Sea level (but not fluctuation through the section) is mentioned briefly in the geological setting and also briefly mentioned in the discussion - p14 line 23-24 "...concluded that TOC enrichment in the Cleveland Basin occurred during times of transgression.." Yet the authors do not provide the context, or critique, of sea level fluctuation as a potential control on the distribution of sedimentary facies and organic matter through the KCF. It is extremely important the authors give more consideration to the role of sea level fluctuation. This should feature in the geological setting and discussion, and perhaps also in the introduction.

- Could the authors comment on sea level during this period? Both in terms of eustatic sea level and local sea level variation. I am assuming no ice caps..?

- Could the authors provide comment on the record of sea level through this interval? Could this alternation reflect sea level fluctation? (I am assuming not, but for completeness..) (page 11, line 2)

Sequence stratigraphic principles have been applied to the coastal outcrops of Kimmeridge Clay Formation at Boulonnais (France) and applied to coeval sediments in the Wessex and Cleveland Basins (Herbin *et al.*, 1995). Organic enrichment does not coincide with fluctuations in sea level so sea level cannot be the primary driver. Furthermore, Powell (2010) states that regional sea level reconstructions in the Cleveland Basin cannot be correlated with the global sea level curve. There is also no evidence for ice at the poles during this time (Dera *et al.*, 2011). We have updated the manuscript to reflect this.

**1.5   Prose and grammar**

The prose should be improved throughout this manuscript. For example, many sentences are too long and/or poorly structured. I have highlighted some specific cases in the annotated PDF. In many cases this can be resolved quickly, by splitting one long sentence into 2 or 3 shorter sentences. This will help communicate the science. In places the language is too informal and mixes tenses. E.g. Page 1, lines 24 and 27.

We kept this in mind while revising the manuscript.

In my opinion these sentences would be better placed after the authors introduce the 'Hadley Cell' hypothesis. The logic of the abstract is then 1) the KCF is highly organic-rich 2) an

expanded Hadley Cell is thought to explain the distribution of organic-rich intervals 3) In order to test this hypothesis, we present a petrographic and geochemical dataset...etc.

We have restructured the abstract to accommodate this.

Should this read 'Stable Isotope Facility?' Or even NEIF? (Please check correct nomenclature with Mel Leng.)

We have changed to the BGS stable isotope facility (part of the National Environmental Isotope Facility.

kyrs not kyr

We prefer kyr (without the 's') as this is used widely in the literature. However, will update should the editorial team request it.

Is it acceptable to include references in the abstract? Perhaps just say "It has been recently postulated..."

We have rewritten the abstract to omit references.

I suggest replace 'energies' terminology throughout this manuscript. Perhaps it is better to say "...most likely due to the action of vigorous bottom currents at seafloor..."

We have included a definition of 'depositional energies' to clarify what we refer to.

I prefer use of 'sediment deposition' rather than sedimentation. Sedimentation means sediment formation rather than deposition, which I think is the authors' meaning here.

We opted to use the Oxford Dictionary definition of *sedimentation*: The process of settling or being deposited as a sediment.

should replace with 'reduced ventilation in bottom waters' or something similar -This is because "increased... redox conditions.. " doesn't make sense

We have updated this in the manuscript.

siliciclastic? or calciclastic? definitions required

The manuscript has been updated to use siliciclastic.

**1.6   Figures**

Palaeolatitude here? (page 19)

We have added palaeolatitude to the figure.

Is 1c sourced from BGS mapping? If so, the authors should check whether  to reference DigMapGB-625 or something similar. Mapping data is copyrighted, but at this scale is likely covered by the Open Government License. Please investigate the source of this map and check the BGS website for referencing protocols if appropriate. (page 19)

This is a figure redrawn from the review paper by Powell (2010).

This ref to 1a is not in sequence. Either re-arrange text so that 1a comes first, or re-arrange fig 1 a-c itself

We have done this in the revised manuscript

But this ms does not include any electron microphotographs in the figs? If this is correct, I suggest the authors remove any mention of SEM, or add example electron microphotographs.
We have added a new figure containing SEM images to the revised manuscript.

If this is scaled, does this corner actually represent 100%? Or perhaps 50%? Might be better to plot CaO (without x2) and then scale the axes appropriately (page 22)
We have used a conventional and widely accepted method of plotting this data (e.g. Brumsack, 1989).

Why not add a legend to this plot? (page 22)
We have included a legend on this plot in the revised manuscript.

Does this delineate a water mass? i.e., perhaps a freshwater cap?) If so, why is the chemocline positioned beneath the pycnocline on b and c?
The green layer represents surface productivity.

Do the authors suggest bottom waters were fully oxic? Or perhaps sub-oxic or ferruginous? If so, perhaps this fig. needs modifying. And do the authors suggest sulphide was completely absent in bottom waters? Is this consistent with the trace element record? (page 28)
We propose that HVMIs are characterised by variations between fully oxic and euxinic conditions.

I think the authors need to put double-ended arrows here and perhaps some comment, to show the chemocline was unstable - i.e., 'particulate shuttle' conditions requires chemocline fluctuation down to seabed. i.e., Fig 5b in Algeo and Tribovillard, 2009, Chemical Geology
This is not correct. The particulate shuttle in the Black Sea, for example, is active despite the chemocline being sometimes thousands of meters above the seabed. The main purpose of the shuttle is to get metals adsorbed to Fe/Mn oxides in the oxic part of the water column, then shuttle them down into the anoxic/euxinic part where they are released and taken up into sulphides or organic matter – and importantly, large parts of the Mn gets re-oxidised at the chemocline so it can continue "pumping" trace metals down into the anoxic/euxinic waters.

Grey and black circles are OK but not the easiest to distinguish - I suggest switching to circles and squares, or circles and crosses, or open circles and filled circles
We have changed the grey circles to grey squares.

**1.7   Samples, sample information, and sample locations**

What is the name of the type section? Where is it located? (the authors could supply coordinates, and/or refer to the BGS memoir for the sheet)
We have included this in the revised manuscript.

Please include mE and mN coordinates for this borehole and projection system (ideally British National Grid), or long & lat with geographic system
We have included this in the revised manuscript.

Please add the ref for this biostrat

We have included this in the revised manuscript.

Colour does not necessarily imply increased TOC - it is also a function of mineralogy, particularly sulphides (pyrite is not necessarily correlated with TOC)

Data from the IFPEN archives report that OC-rich intervals are darker in colour, thus in this study it was appropriate to consider this observation during high resolution sampling.

---

## Referee Comment (RC2) · Jennifer McKay (Referee) · 15 May 2019

This manuscript uses a wide variety of geochemical proxies, which they integrate with sedimentological information, to investigate the depositional environment of the Kimmeridge Clay Formation in the Cleveland Basin. In general, the conclusions they reach are valid and after some moderate revisions I recommend publication of this manuscript.

Important revisions:

1. Section 3 (Materials and methods) should include information about precision and

accuracy, as well as better descriptions of the analytical methods; notably for the analysis of carbon isotopes. Details are provide below.

2. The manuscript is a bit disorganized in places (e.g., interpretations more suited to the discussion are found within the result section, figures are out of order). Details are provided below.

3. The discussion of the d13Corg data is limited and lacking in detail. The authors simply say lighter values indicate more terrestrial organic matter. While this is correct they do not providing references / background information to support this interpretation. In general, Section 5.2 is lacking in appropriate references.

Specific Comments:

Page 1, Line 27 – You talk about "three states that produced a distinct cyclicity" however the paper is primarily divided into two units LVMIs and HVMIs. This is a bit confusing.

Page 3, Lines 27 and 28 – Why are "ocean overturn, salinity/temperature stratification and redox conditions" mentioned in this sentence about organic carbon enrichment? Something does not make sense here.

Page 4, line 13 – The sediments might be thermally immature but 425°C is high and would have undoubtedly affected the sediments. Diagenetic alteration can occur at temperatures below 100°C. Just something to keep in mind especially when looking at the Hg data.

Section 3, General comment – Where is the information about precision and accuracy of the geochemical analyses?

Page 4, line 20 – It is highly unlikely that sedimentation was linear throughout this time period. The changes in climate and resulting changes in sedimentation were simply to drastic. It would be better to use the biostratigraphy to estimate how much time this covers.

[Figure]

Page 4, line 28 – XRF measures major and minor (not trace) elements. For consistency, you should also list which elements were analyzed

Page 4, lines 28 to 29 – More information about how d13Corg was measured is needed. What equipment was used? Were the samples pretreated to remove carbonate before d13C was analyzed? What standards were analyzed?

Page 5, lines 9 to 15 – This discussion about Hg does not belong here. Move to page 8 (First paragraph of the discussion).

Page 5, line 17 – You need to provide more detail on how the boundaries of the HVMIs were determined. I'm requesting this because the lowermost zone is thicker than I think it should be based on carbonate and d13Corg values.

Page 6, lines 2 and 3 – Did you measure grain sized? If not, how are you determining the difference between "medium to coarse mud-size" and "fine to coarse mud-sized"? Also, there is technically no such thing as a "mud-sized grain". Grain size is subdivided into clay, silt and sand . . . mud is a mix of clay and silt, with perhaps a bit of fine sand.

Page 6, lines 11 and 12 – According to what you wrote at the end of page 5 LVMIs are composed of two facies (1 and 6) so facies 4 should be discussed in Section 4.1.2

Page 6, line 16 – What three samples are you referring to in Figure 2? I assume it is those samples identified by arrows but you should make this clear by saying (Fig 2, arrows in CaCO3 plot).

Page 6, lines 20 to 30 – You need to emphasize the differences between Facies 2 and 3. It would help if you referred to Figure 3 more often throughout Section 4.1.

Pages 7 and 8 – Section 4.2 should simply describe the results. For example, LVMIs have lower d13Corg values than the HVMIs (Fig. 2). The interpretation of the d13C data should not occur until the Discussion. Furthermore, a lot of the results are not mentioned (e.g., CaCO3 content in the LVMIs). This section needs to be cleaned up.

Page 7, lines 18 to 20 – This discussion about the d13Corg data needs to move to Section 5. Once you do this, please provide references for this statement "Marine organic matter has a higher d13Corg than terrestrial OM".

Page 8, line 24 – You need to better explain how Hg/TOC data rules out a volcanogenic sediment source. Also, these sediments got rather warm during burial (425°C) . . . would this affect the Hg content?

General comment about Discussion – Section 5.1 merges a lot of information, much of which you haven't discussed yet (e.g., the d13Corg data which is, or should be, discussed in section 5.2). I suggest you swap the order of sections 5.1 and 5.2 (i.e., discuss productivity and organic carbon first).

Page 8, line 30 – You cannot identify what clay minerals are present (i.e., kaolinite vs illite) using just petrography. This requires specialized X-ray diffraction techniques.

Page 10, Section 5.2 – This section might be easier to understand if you divided into subsections as you did for Section 5.3 (i.e., a subsection for LVMIs and HMVIs).

Page 10, line 7 – A brief discussion about what type II and III organic matter is would be helpful. Also, this is the place to discuss how d13Corg can be used to identify terrestrial vs marine organic matter. Can you estimate the end member marine and terrestrial values? If yes, you could estimate the fraction of organic matter that is terrestrial vs marine. I assume you do not have %N data but if you did you could also use Corg/N ratios . . . although not all terrestrial organic matter has high Corg/N ratios.

Page 10, line 11 – So, if the LVMIs are deposited in a more "distal" location relative to the HMVIs . . .. why is terrestrial organic matter higher in the LVMIs?

Page 10, line 23 – It is extremely unlikely that the euphotic zone reached the seafloor if waters depths were greater than 200 meters, which is the approximate limit of the euphotic zone in the modern ocean. That said, light limitation would only be an issue in at higher latitudes (i.e., mid-latitudes to polar regions) where in winter it limits productivity

along with strong storm mixing. Nutrient supply is undoubtedly controlling productivity.

Page 11, lines 1 and 2 – I agree that nutrient availability may have influenced whether the HVMI sediments were TOC-rich or carbonate-rich. In the modern ocean a well-stratified water column, characterized by a stable, warm, nutrient-poor surface layer does favor coccolithophorids over diatoms.

Page 11, line 4 – The discussion about fecal pellets and zooplankton is not critical to the discussion and perhaps distracting. I suggest removing it. If you prefer to keep it then write it in the same way as it is written on page 17and include some references that support the idea that packaging of organic matter into fecal pellets speeds the delivery to the seafloor and aids in preservation. Also, zooplankton were probably around during the formation of both LVMIs and HVMIs . . . but bioturbation of the LVMIs destroyed the fecal pellets in these units. Interestingly, in modern sediments fecal pellets tend to be preserved where you have sediment starvation regardless of whether there is bioturbation or not.

Page 11, lines 12 and 13 – In section 4.2.1 (page 7, line 25) you mention that trace metal enrichments in LVMIs are low (i.e., around 1) which agrees with metal/Al ratios shown in Figure 6. However, here you say there are trace metal enrichments and imply they are large. What I believe you mean to say is that when you look at Mo and U EFs (Fig 8 . . . which should be Fig 7) there is some enrichment indicating the development of reducing conditions after burial. Please clarify.

Figures 7 and 8 – you need to switch the order of Figures 7 and 8 so they match the order they are discussed in the text. Make sure to change the figure numbers in the text as well. Also, the figure caption for Fig 8 (what will be Fig 7) has some errors (e.g., the panels that depict the HVMIs are blue not grey).

Page 11, lines 16 to 24 – this discussion about TOC should be moved to Section 5.2 (i.e., the Productivity and organic matter composition).

Page 11, line 25 - I think you mean HVMIs . . . not LVMIs in the title of section 5.3.2

Page 11, line 28 – technically anoxic refers to porewaters with sulfide and euxinic to a water column (i.e., bottom waters) with sulfide but I know there is some "debate" about these terms.

Page 12, lines 4 and 5 – You still have not discussed Figure 7 and you are now mentioning Figure 9. I would remove this sentence and wait until Section 5.3.4 to discuss it.

Page 12, lines 6 to 9 – I agree, what you are seeing in the KCF looks very similar to the Baltic Sea.

Page 12, line 27 – Do you mean that the high As/Al and Sb/Al ratios within the HVMIs support the idea of higher pyritisation? A better way of saying this is that pyritization probably explains the enrichment of metals known to accumulate with pyrite (e.g., As, Sb and Mo).

Page 13, line 12 to 13 – You need to emphasize that those steep geochemical gradients are driven by changes in oxygen vs sulfide (i.e., redox).

Page 13, line 16 – replace the word "phases" with "particulates" at both the start and end of this line.

Page 13, line 24 – You need to explain the Mn better (e.g., what about the Mn carbonates). I suggest removing this sentence and adding a paragraph about Mn at the end of this section.

Page 14, lines 4 and 5 – remove "with associated enrichments of redox sensitive/sulfide forming trace metals (Fig. 6)" . . . it is not needed here.

Page 14, lines 6 to 8 – Also remove this sentence. It does not read well and you basically say the same thing in the next paragraph. So, now you can merge the second and third paragraphs on this page.

Page 14, line 19 – Can you provide some Mo/U enrichment ratio data for other rock and sediments (not just the Tribovillard et al 2012 data which is also the KCF) to support your conclusion that the Mn-Fe shuttle must have been working during HVMI deposition. A figure similar to your Figure 9 would work. If there are not enough data to do this, you need to say "probably facilitated".

Page 14, lines 26 and 27 – I'm not sure what you mean by "which is in good agreement with our Fe/Al and Mn/Al data discussed above". I'd remove this.

Page 14, line 28 – didn't you say sediment redox conditions in the LVMIs become anoxic (Page 11, line 14)? It cannot be both.

Page 15, line 5 – no need to reference Brumsack (2006) here.

Page 15, line 6 – average shale or UCC?

Page 15, line 15 – I thought that Mn (in the form of carbonate) was enriched in the HVMIs (Page 12, line 1). Perhaps depletion is not the correct word or you need to clarify "depleted in comparison to . . .".

Page 15, line 25 – by "the two intervals of deposition" are you referring to the Black Sea and HVMIs? This is unclear.

Page 16, line 4 – What do you mean by "If at all. . ."? Are you trying to suggest the Baltic is the best modern analogue we have? Right, I see where you say this on line 16. Perhaps you need to move this last paragraph (lines 16 to 21) up to line 4.

Page 16, lines 23 to 16 – You repeat yourself a couple of times. This paragraph needs to be rewritten or just remove it.

Page 16, line 18 – rewrite as follows ". . .fluctuated between LVMI and HVMI deposition."

Page 16, line 10 – I thought you said high TOC concentrations related to the fact it was terrestrial. Where did you say that reducing conditions lead to TOC enrichment in the LVMIs?

Page 17, general comments – do fluctuations in sea level not play a role in the changes we are seeing?

Page 17, line 4 – write as follows "an increase in OM flux rate to the seafloor"

Page 17, line 7 – you mention elevated depositional energies during the during oxygenation events associated with HVMIs but in Figure 3 the erosional features are associated with Facies 2, which I though was the "anoxic" facies. Please clarify.

Page 17, line 17 – you imply that both the LVMI and HVMI units ("the studied section") are similar to what you see in the Baltic Sea. Is this what you mean to say?

Page 17, lines 19 to 24 – you discuss the timescale and mechanism that is mostly likely controlling the change between LVMIs and HVMIs. What about within the HVMIs? These changes must be much more rapid. Any educated guesses about what might be controlling this?

Page 18, line 5 – What data repository will you use?

Spelling/Grammar:

Page 3, Line 24 – the word basins should not be capitalized Page 4, Line 1 – add the word "well" before "exposed" Page 4, line 9 – need a space between core list and (Fig. 1c) Page 7, line 25 (also on page 8, line 15) – add the word "relative" before "to UCC" Page 7, lines 29 and 30 – The value -27.0 should come before -28.8 ‰ Page 12, lines 1 to 9 – some grammar issues in this paragraph Page 12, line 32 – replace "of" with "for" (i.e., the need for caution when using Mo. . .) Page 13, line 22 – proxies not proxy Page 16, line 14 – replace "is" with "are" and spell out Figure (e.g., in the Baltic Sea are plotted on Figure 9.). Page 16, line 23 – remove the "an" before "insight"

---

## Author Response (AR1)

**Reply to Reviewer 1**

We thank Professor Algeo for his time and providing helpful comments. We have replied to each point in the review here and will submit a revised manuscript that combines comments from all reviewers once all reviews have been completed. Please find our point to point reply below. Text from the review is in blue and our replies are in black.

Overall, this study is very well-executed, and the interpretations and conclusions are quite reasonable. It should become acceptable for publication following minor revision. I offer some insights on a few important issues as well as some minor comments that the authors should consider during the revision stage.

The approach adopted in this study is to be commended in one respect in particular: It is useful to identify the main components of a sample set, and to link the geochemistry of the bulk samples to these components. This should be self-evident, but too frequently geochemical proxies are interpreted in chemostratigraphic studies with little consideration given to the underlying host sediment fractions.

Thank you. A key aim of this study was to use petrographic information to support geochemical interpretation.

Major issues:
1) Flux calculations should be presented for arguments that invoke sediment fluxes. For example: Page 8, line 31: "The fine-grained nature of the sedimentary rock may indicate a siliciclastic starvation process." When making claims regarding high or low siliciclastic or organic fluxes, it is generally a good idea to support them with some actual flux calculations. Making reliable flux calculations may be difficult for poorly dated formations, but the present study units are exceedingly well-dated, spanning the Pectinatites wheatleyensis to Pectinatites pectinatus ammonite biozones. This interval corresponds to 1.5 Myr (ca. 151.2-149.7 Ma) per the 2012 Geologic Time Scale (Gradstein et al., 2012, chapter 26). The study interval comprises 45 m, so its average sedimentation rate is 30 m/Myr–in other words, a pretty average cratonic rate. There might be condensed intervals within this succession, but it is not sediment-starved as a whole, as implied by the statement above.

We fully agree that flux rates are easy to include and add value to the manuscript. We have incorporated flux rates into the revised manuscript and have reworded the text to remove the siliciclastic starvation reference.

Page 9, line 8: "the occurrence of normally graded beds with erosional bases in TOC- rich sections of the HVMIs indicates an energetically dynamic setting." A combination of high energy levels and sediment starvation would produce a lag deposit, i.e., concentrated high-density and/or resistant clasts such as fossil, pyrite, and/or phosphate grains. Are there any features of this type in the study succession?

We do not see evidence of lag deposits. Given the calculation of the flux rates, it is unlikely that the system was sediment starved so this is not unexpected. We have added another figure to the revised manuscript containing optical light and SEM images to further illustrate our sedimentary descriptions.

Page 9, line 31: "Based on the chronostratigraphic time frame for the Yorkshire and Dorset sections (Armstrong et al., 2016; Huang et al., 2010) . . ." If there is a published astrochronology for the study formations (Huang et al., 2010), why not integrate it into the present study and discuss its implications for the duration and accumulation rates of these formations?

We agree that an astrochronological framework would be useful in investigating different flux rates; however, the existing framework is based on a laterally (more or less) equivalent section and assumes continuous sedimentation. We have petrographic evidence that there is sediment missing from the section (e.g. erosional surfaces). Furthermore, we do not have enough data for a statistically robust examination of astrochonological cycles in the Yorkshire section (i.e., the section is too short and does not contain enough cycles). Therefore, we feel transferring the astrochronological framework from the Dorset section to the Yorkshire section introduces large uncertainties. Nevertheless, we have included the framework for reference and have discussed the strengths and weaknesses of this approach in the revised manuscript.

2) Interpretation of controls on high TOC or high CaCO3 intervals:
Page 10, line 24: "We therefore propose that nutrient availability was the likely driver of changes in productivity. . . . The wet-dry cycles proposed by recent climate modelling (Armstrong et al., 2016) may therefore be the key driver behind oscillations in the production and preservation of TOC, i.e. the switching between the LVMIs and HVMIs." This is certainly possible but might be difficult to prove. An alternative hypothesis is that organic productivity and sinking fluxes were held more-or-less constant, and the large variations in TOC content were driven by variable influx of siliciclastics, leading to variable dilution of the organic carbon flux. Perhaps the authors could provide arguments countering this alternative hypothesis? Of course, it is also possible that both nutrient and siliciclastic fluxes were covarying in tandem.
We agree that this is an alternative hypothesis but do not have the data to distinguish between the two. We have updated the manuscript to reflect this.

Here is where the astrochronology of the study formations might help–if a characteristic periodic signal (e.g., 100-kyr eccentricity cycles) is present, then it is potentially possible to calculate short-term variations in sedimentation rates in a study section, rather than being limited to an average sedimentation rate for the entire section (as calculated above). An example of application of a floating time scale to analysis of short-term sedimentation rate variation is given in Algeo et al. 2011 (Algeo, T.J., Kuwa- hara, K., Sano, H., Bates, S., Lyons, T., Elswick, E., Hinnov, L., Ellwood, B., Moser, J. and Maynard, J.B., 2011. Spatial variation in sediment fluxes, redox conditions, and productivity in the Permian–Triassic Panthalassic Ocean. Palaeogeography, Palaeoclimatology, Palaeoecology, 308(1-2), pp. 65-83.).
The astrochronological framework was built upon the laterally equivalent type section in Dorset and assumes continuous sedimentation, therefore we do not feel it would be appropriate to calculate rates of short-term variation using astrochronology from a different section.

Page 10, line 30: "Carbonate productivity, mainly in the form of coccoliths, varies throughout the studied KCF section and is at its maximum within the carbonate-rich sections of the HVMIs." Probably correct, but again this represents an assumption, not a proven fact, and one could argue (as for TOC variations; see above) that variable siliciclastic influence controlled variations of carbonate content in the study section.
We agree and have included alternative hypothesis in the revised manuscript.

Page 11, line 13: "However, enrichment factors of redox sensitive trace elements (Mo and U; Fig. 8) … indicate that during the deposition of the LVMIs, the sediment pore water was suboxic to anoxic." This is open to interpretation, but my opinion is that the modest Mo-EFs (mean 4, max 20) and U-EFs (mean 2, max 4) of the LVMIs indicate overwhelmingly suboxic conditions. These values are not strongly supportive of anoxic conditions.
We agree. We have updated the manuscript accordingly.

Section 5.3.4: One point about particulate shuttles that is not made clearly here is that they seem to be most effective at authigenic trace metal enrichment when redox conditions fluctuate strongly, as opposed to stably euxinic conditions (see Algeo and Tribovillard, 2009).
We have updated the manuscript accordingly.

3) Minor issues:
Page 9, line 14: "Owing to the shallow gradients and vast extent of epicontinental seaways, sediment dispersal in the LVMIs, which are dominated by terrigenous mud, is likely to have been controlled by wind- and tide-induced bottom currents (Schieber, 2016)." Whether winds and tides could induce significant bottom currents would depend on water depths–at tens of meters, they would be important but at hundreds of meters much less so. The geologic background section (page 3, line 30) indicates considerable uncertainty regarding water depths in the NW European Sea, so the potential influence of bottom currents is uncertain.
We have updated the manuscript to reflect this.

Page 10, line 5: "Biological components (coccolithophores, foraminiferans, and organic carbon) occur in differing proportions throughout the section (Figs. 2 and 3). Our petrographic observations (Fig. 2)…" The presented petrographic data appear to be entirely visual/descriptive. Why not undertake point counts of organic maceral types? This would be provide more quantitative information about the nature of the organic fraction that could be compared with other data (e.g., d13C-org)
Given the fine-grained nature of the sediment, point counting these samples would have to be done under SEM, where organic matter type could not be determined. Instead, we have further and more fully integrated published RockEval data that provides information on organic matter type.

Page 10, line 22: "Water depth is not likely to have exceeded a few hundreds of meters in the distal Cleveland Basin (Bradshaw et al., 1992), suggesting that the euphotic zone could have reached the seafloor and light did not limit primary productivity." This statement betrays an incomplete understanding of the photic zone. Light intensity is attenuated quickly and drops to 30% of surface levels by 10 m and to a few percent by 100 m water depth in clear water; in turbid water, the rate of attenuation can be much faster with depth. Most primary productivity is typically in the upper 10 m of the water column, and there will be very little productivity at the depths suggested here.
We have adjusted the text to reflect this in the revised manuscript.

Page 14, line 32: "we can confirm that the repeated development of anoxic/euxinic conditions in the distal Cleveland Basin was most likely due to high primary productivity, and possibly salinity stratification due to high amounts of freshwater runoff". We encourage the authors to investigate the use of paleosalinity proxies to evaluate changes in freshwater runoff in this depositional system. Check this paper for paleosalinity analysis techniques:
Wei, W., Algeo, T.J., Lu, Y., Lu, Y., Liu, H., Zhang, S., Peng, L., Zhang, J. and Chen, L., 2018. Identifying marine incursions into the Paleogene Bohai Bay Basin lake system in northeastern China. International Journal of Coal Geology, 200, pp. 1-17.
We have looked at the suggested paleosalinity proxies and they do not show systematic variations, most likely indicating that the system was marine at all times.

Page 15, line 14: "The HVMIs in the present study bear similarities to the Gulf of California in that they exhibit similarities in Cd enrichment and Mn depletion". The 2016 study by Tim Sweere is highly relevant in this regard and should be cited:
Sweere, T., van den Boorn, S., Dickson, A.J. and Reichart, G.J., 2016. Definition of new trace-metal proxies for the controls on organic matter enrichment in marine sediments based on Mn, Co, Mo and Cd concentrations. Chemical Geology, 441, pp.235-245.
This reference has been added to the manuscript.

Page 16, line 2: "While the studied interval shares similarities and differences with both upwelling and anoxic basin type settings, we are still lacking an appropriate modern analogue. Palaeogeography exerts a fundamental control on sedimentation, in particular, TOC enrichment, but there is no modern-day example of a shallow epicontinental seaway." Agreed, but the authors should consider the examples provided in Algeo et al. (2008):
Algeo, Thomas J., Philip H. Heckel, J. Barry Maynard, Ronald C. Blakey, Harry Rowe, B. R. Pratt, and C. Holmden. "Modern and ancient epeiric seas and the super-estuarine circulation model of marine anoxia." Dynamics of Epeiric Seas: Sedimentological, Paleontological and Geochemical Perspectives: Geological Association Canada Special Paper 48 (2008): 7-38.
This reference has been added to the manuscript

**Reply to Reviewer 2**

We thank Dr McKay for her time and detailed review, which has certainly improved our manuscript. Please find our point-to-point reply below. Text from the review is in blue and our replies are in black.

This manuscript uses a wide variety of geochemical proxies, which they integrate with sedimentological information, to investigate the depositional environment of the Kimmeridge Clay Formation in the Cleveland Basin. In general, the conclusions they reach are valid and after some moderate revisions I recommend publication of this manuscript.

Important revisions:
1.      Section 3 (Materials and methods) should include information about precision and accuracy, as well as better descriptions of the analytical methods; notably for the analysis of carbon isotopes. Details are provide below.
2.      The manuscript is a bit disorganized in places (e.g., interpretations more suited to the discussion are found within the result section, figures are out of order). Details are provided below.
3.      The discussion of the d13Corg data is limited and lacking in detail. The authors simply say lighter values indicate more terrestrial organic matter. While this is correct they do not providing references / background information to support this interpretation. In general, Section 5.2 is lacking in appropriate references.

The important revisions listed above are expanded upon in specific comments section so they are addressed below.

Specific Comments:

Page 1, Line 27 – You talk about "three states that produced a distinct cyclicity" however the paper is primarily divided into two units LVMIs and HVMIs. This is a bit confusing.
We have rephrased this to explain that that HVMIs comprise carbonate-rich and organic carbon-rich units.

Page 3, Lines 27 and 28 – Why are "ocean overturn, salinity/temperature stratification and redox conditions" mentioned in this sentence about organic carbon enrichment? Something does not make sense here.
We have reworded this to make it clearer.

Page 4, line 13 – The sediments might be thermally immature but 425°C is high and would have undoubtedly affected the sediments. Diagenetic alteration can occur at temperatures below 100°C. Just something to keep in mind especially when looking at the Hg data.
We agree. We have added a sentence to consider this. Ongoing research suggests thermal maturity would affect Hg contents depending upon which sedimentary component hosts the Hg. i.e. if the Hg resides in the pyrite, it may be less susceptible to diagenetic alteration (Them et al., 2019).

T.R. Them, C.H. Jagoe, A.H. Caruthers, B.C. Gill, S.E. Grasby, D.R. Gröcke, R. Yin, J.D. Owens, Terrestrial sources as the primary delivery mechanism of mercury to the oceans across the Toarcian Oceanic Anoxic Event (Early Jurassic), Earth and Planetary Science Letters, Volume 507, 2019, Pages 62-72, ISSN 0012-821X, https://doi.org/10.1016/j.epsl.2018.11.029.

Section 3, General comment – Where is the information about precision and accuracy of the geochemical analyses?
We have added this to Supplement A.

Page 4, line 20 – It is highly unlikely that sedimentation was linear throughout this time period. The changes in climate and resulting changes in sedimentation were simply too drastic. It would be better to use the biostratigraphy to estimate how much time this covers.
Biostratigraphy has been used to calculate sedimentation rate. We have expanded upon the calculation of linear sedimentation rates and the associated problems.

Page 4, line 28 – XRF measures major and minor (not trace) elements. For consistency, you should also list which elements were analysed
We have listed the elements that were analysed in the revised manuscript.

Page 4, lines 28 to 29 – More information about how d13Corg was measured is needed. What equipment was used? Were the samples pretreated to remove carbonate before d13C was analyzed? What standards were analyzed?
This information is available in Supplement A, but we have moved it into the manuscript for clarity.

Page 5, lines 9 to 15 – This discussion about Hg does not belong here. Move to page 8 (First paragraph of the discussion).
We have made this revision.

Page 5, line 17 – You need to provide more detail on how the boundaries of the HVMIs were determined. I'm requesting this because the lowermost zone is thicker than I think it should be based on carbonate and d13Corg values.

The HVMIs are defined as intervals in which carbonate and organic carbon contents are highly variable, thus the lowermost HVMI is the correct thickness as it incorporates the exceptionally high TOC values. We have added a sentence to clarify this in the manuscript.

Page 6, lines 2 and 3 – Did you measure grain sized? If not, how are you determining the difference between "medium to coarse mud-size" and "fine to coarse mud-sized"? Also, there is technically no such thing as a "mud-sized grain". Grain size is subdivided into clay, silt and sand . . . mud is a mix of clay and silt, with perhaps a bit of fine sand.

Grain size was measured on the SEM using a ruler in the AZTEC software (used to run the SEM). We choose to use the Lazar et al 2015 classification system where sand is defined as any grain between 62.5 μm and 2000 μm, coarse mud is between 62.5 μm and 32 μm, medium mud is between 32 μm and 8 μm and fine mud is anything less than 8 μm. See Lazar et al 2015 for detailed discussion. We have added a sentence to the manuscript to clarify this.

Page 6, lines 11 and 12 – According to what you wrote at the end of page 5 LVMIs are composed of two facies (1 and 6) so facies 4 should be discussed in Section 4.1.2

We have moved the discussion of facies 4 to Section 4.1.2.

Page 6, line 16 – What three samples are you referring to in Figure 2? I assume it is those samples identified by arrows but you should make this clear by saying (Fig 2, arrows in CaCO3 plot).

We have made this revision.

Page 6, lines 20 to 30 – You need to emphasize the differences between Facies 2 and 3. It would help if you referred to Figure 3 more often throughout Section 4.1.

We have made this revision.

Pages 7 and 8 – Section 4.2 should simply describe the results. For example, LVMIs have lower d13Corg values than the HVMIs (Fig. 2). The interpretation of the d13C data should not occur until the Discussion. Furthermore, a lot of the results are not mentioned (e.g., CaCO3 content in the LVMIs). This section needs to be cleaned up.

We have rewritten this section to address this comment.

Page 7, lines 18 to 20 – This discussion about the d13Corg data needs to move to Section 5. Once you do this, please provide references for this statement "Marine organic matter has a higher d13Corg than terrestrial OM".

We have made this revision in the revised manuscript and an added appropriate reference to support the statement.

Page 8, line 24 – You need to better explain how Hg/TOC data rules out a volcanogenic sediment source. Also, these sediments got rather warm during burial (425∘C) ... would this affect the Hg content?

We have expanded upon the explanation of Hg/TOC as a proxy and commented on the expected effect of thermal maturity on Hg contents.

General comment about Discussion – Section 5.1 merges a lot of information, much of which you haven't discussed yet (e.g., the d13Corg data which is, or should be, discussed in section 5.2). I suggest you swap the order of sections 5.1 and 5.2 (i.e., discuss productivity and organic carbon first).
We agree that this would improve clarity so have made this change in the revised manuscript.

Page 8, line 30 – You cannot identify what clay minerals are present (i.e., kaolinite vs illite) using just petrography. This requires specialized X-ray diffraction techniques.
Low resolution XRD data was published from the Ebberston 87 Core by Herbin et al. (1991). We have included this work in the revised manuscript.

Page 10, Section 5.2 – This section might be easier to understand if you divided into subsections as you did for Section 5.3 (i.e., a subsection for LVMIs and HMVIs).
We agree that this improves clarity so have made this revision.

Page 10, line 7 – A brief discussion about what type II and III organic matter is would be helpful. Also, this is the place to discuss how $\partial13Corg$ can be used to identify terrestrial vs marine organic matter. Can you estimate the end member marine and terrestrial values? If yes, you could estimate the fraction of organic matter that is terrestrial vs marine. I assume you do not have %N data but if you did you could also use Corg/N ratios . . . although not all terrestrial organic matter has high Corg/N ratios.
We have expanded upon the discussion on use of $\partial13Corg$ as a proxy for organic matter type.

Page 10, line 11 – So, if the LVMIs are deposited in a more "distal" location relative to the HMVIs . . .. why is terrestrial organic matter higher in the LVMIs?
Given the epicontinental seaway setting, we have removed the terms 'proximal' and 'distal' from the manuscript as they are unlikely to be very apparent in the fine-grained sedimentary record (owing to the shallow gradients).

Page 10, line 23 – It is extremely unlikely that the euphotic zone reached the seafloor if waters depths were greater than 200 meters, which is the approximate limit of the euphotic zone in the modern ocean. That said, light limitation would only be an issue in at higher latitudes (i.e., mid-latitudes to polar regions) where in winter it limits productivity along with strong storm mixing. Nutrient supply is undoubtedly controlling productivity.
We agree and have revised that paragraph to reflect this.

Page 11, lines 1 and 2 – I agree that nutrient availability may have influenced whether the HVMI sediments were TOC-rich or carbonate-rich. In the modern ocean, a well- stratified water column, characterized by a stable, warm, nutrient-poor surface layer does favor coccolithophorids over diatoms.
We are pleased that the reviewer supports our interpretation.

Page 11, line 4 – The discussion about fecal pellets and zooplankton is not critical to the discussion and perhaps distracting. I suggest removing it. If you prefer to keep it then write it in the same way as it is written on page 17and include some references that support the idea that packaging of organic matter into fecal pellets speeds the delivery to the seafloor and aids in preservation. Also, zooplankton were probably around during the formation of both LVMIs and HVMIs . . . but bioturbation of the LVMIs destroyed the

fecal pellets in these units. Interestingly, in modern sediments fecal pellets tend to be preserved where you have sediment starvation regardless of whether there is bioturbation or not.
We have added appropriate references to support our statement.

Page 11, lines 12 and 13 – In section 4.2.1 (page 7, line 25) you mention that trace metal enrichments in LVMIs are low (i.e., around 1) which agrees with metal/Al ratios shown in Figure 6. However, here you say there are trace metal enrichments and imply they are large. What I believe you mean to say is that when you look at Mo and U EFs (Fig 8 . . . which should be Fig 7) there is some enrichment indicating the development of reducing conditions after burial. Please clarify.
We have corrected this in the revised manuscript.

Figures 7 and 8 – you need to switch the order of Figures 7 and 8 so they match the order they are discussed in the text. Make sure to change the figure numbers in the text as well. Also, the figure caption for Fig 8 (what will be Fig 7) has some errors (e.g., the panels that depict the HVMIs are blue not grey).
We have made this revision.

Page 11, lines 16 to 24 – this discussion about TOC should be moved to Section 5.2 (i.e., the Productivity and organic matter composition).
We have made this revision.

Page 11, line 25 - I think you mean HVMIs . . . not LVMIs in the title of section 5.3.2
We have made this revision.

Page 11, line 28 – technically anoxic refers to porewaters with sulfide and euxinic to a water column (i.e., bottom waters) with sulfide but I know there is some "debate" about these terms.
We define 'anoxic' as water depleted of oxygen and 'euxinic' as water depleted of oxygen and with free H2S, these definitions are irrespective of depth.

Page 12, lines 4 and 5 – You still have not discussed Figure 7 and you are now mentioning Figure 9. I would remove this sentence and wait until Section 5.3.4 to discuss it.
We agree this improves the manuscript and have made this revision in the manuscript.

Page 12, lines 6 to 9 – I agree, what you are seeing in the KCF looks very similar to the Baltic Sea.
We are pleased that the reviewer supports our interpretation.

Page 12, line 27 – Do you mean that the high As/Al and Sb/Al ratios within the HVMIs support the idea of higher pyritisation? A better way of saying this is that pyritization probably explains the enrichment of metals known to accumulate with pyrite (e.g., As, Sb and Mo).
We do mean this so have reworded this sentence to the suggestion made.

Page 13, line 12 to 13 – You need to emphasize that those steep geochemical gradients are driven by changes in oxygen vs sulfide (i.e., redox).
We agree and have done this in the revised manuscript.

Page 13, line 16 – replace the word "phases" with "particulates" at both the start and end of this line.

We have done this in the revised manuscript.

Page 13, line 24 – You need to explain the Mn better (e.g., what about the Mn carbonates). I suggest removing this sentence and adding a paragraph about Mn at the end of this section.
We have done this in the revised manuscript.

Page 14, lines 4 and 5 – remove "with associated enrichments of redox sensitive/sulfide forming trace metals (Fig. 6)" . . . it is not needed here.
We have done this in the revised manuscript.

Page 14, lines 6 to 8 – Also remove this sentence. It does not read well and you basically say the same thing in the next paragraph. So, now you can merge the second and third paragraphs on this page.
We have done this in the revised manuscript.

Page 14, line 19 – Can you provide some Mo/U enrichment ratio data for other rock and sediments (not just the Tribovillard et al 2012 data which is also the KCF) to support your conclusion that the Mn-Fe shuttle must have been working during HVMI deposition. A figure similar to your Figure 9 would work. If there are not enough data to do this, you need to say "probably facilitated".
The Mn-Fe shuttle is identified in a number of deposits besides the KCF so we will mention this and provide references in the revised manuscript.

Page 14, lines 26 and 27 – I'm not sure what you mean by "which is in good agreement with our Fe/Al and Mn/Al data discussed above". I'd remove this.
We have done this in the revised manuscript.

Page 14, line 28 – didn't you say sediment redox conditions in the LVMIs become anoxic (Page 11, line 14)? It cannot be both.
We have removed 'or anoxic' from Page 11, line 14.

Page 15, line 5 – no need to reference Brumsack (2006) here.
We have removed this reference.

Page 15, line 6 – average shale or UCC?
In section 5.3.5, enrichment factors are calculated to average shale to facilitate direct comparison with the Brumsack (2006) compilation.

Page 15, line 15 – I thought that Mn (in the form of carbonate) was enriched in the HVMIs (Page 12, line 1). Perhaps depletion is not the correct word or you need to clarify "depleted in comparison to . . .".
We have replaced 'depletion' with 'depleted in comparison to the HVMIs'.

Page 15, line 25 – by "the two intervals of deposition" are you referring to the Black Sea and HVMIs? This is unclear.
We do and have reworded to clarify this point.

Page 16, line 4 – What do you mean by "If at all..."? Are you trying to suggest the Baltic is the best modern analogue we have? Right, I see where you say this on line 16. Perhaps you need to move this last paragraph (lines 16 to 21) up to line 4.
We have done this in the revised manuscript.

Page 16, lines 23 to 16 – You repeat yourself a couple of times. This paragraph needs to be rewritten or just remove it.
We have rewritten to remove the repetition.

Page 16, line 18 – rewrite as follows ". . .fluctuated between LVMI and HVMI deposition."
We have done this in the revised manuscript.

Page 16, line 10 – I thought you said high TOC concentrations related to the fact it was terrestrial. Where did you say that reducing conditions lead to TOC enrichment in the LVMIs?
TOC in the LVMI is predominantly terrestrial organic matter, the preservation of which is facilitated by suboxic sediment-water interface conditions.

Page 17, general comments – do fluctuations in sea level not play a role in the changes we are seeing?
Previous work (e.g. Herbin et al., 1993) indicates that sea level is not the primary control on organic carbon enrichment in the studied interval.

Page 17, line 4 – write as follows "an increase in OM flux rate to the seafloor"
We have done this in the revised manuscript.

Page 17, line 7 – you mention elevated depositional energies during the during oxygenation events associated with HVMIs but in Figure 3 the erosional features are associated with Facies 2, which I though was the "anoxic" facies. Please clarify.
The different scales of petrography and bulk rock geochemical analyses means that it is impossible to perfectly match the two together. We mean to say that the HVMI, as a whole, have periodically higher depositional energies related to the LVMIs. We have updated the revised manuscript to clarify this.

Page 17, line 17 – you imply that both the LVMI and HVMI units ("the studied section") are similar to what you see in the Baltic Sea. Is this what you mean to say?
We mean to say that the system as a whole, i.e. periodic reoxygenation events, are similar to those observed in the Baltic Sea.

Page 17, lines 19 to 24 – you discuss the timescale and mechanism that is mostly likely controlling the change between LVMIs and HVMIs. What about within the HVMIs? These changes must be much more rapid. Any educated guesses about what might be controlling this?
The resolution of data points within the HVMIs is too low to support any statement about processes operating at these timescales. As such, a higher resolution study would be needed to ascertain these scales but would be difficult owing to the limitations of ammonite biostratigraphy.

Page 18, line 5 – What data repository will you use?
The data will be deposited in the Pangaea repository.

Spelling/Grammar:

Page 3, Line 24 – the word basins should not be capitalized Page 4, Line 1 – add the word "well" before "exposed" Page 4, line 9 – need a space between core list and (Fig. 1c) Page 7, line 25 (also on page 8, line 15) – add the word "relative" before "to UCC" Page 7, lines 29 and 30 – The value -27.0 should come before -28.8 ‰ Page 12, lines 1 to 9 – some grammar issues in this paragraph Page 12, line 32 – replace "of" with "for" (i.e., the need for caution when using Mo. . .)

Page 13, line 22 – proxies not proxy Page 16, line 14 – replace "is" with "are" and spell out Figure (e.g., in the Baltic Sea are plotted on Figure 9.). Page 16, line 23 – remove the "an" before "insight

We have made these corrections in the revised manuscript

**Dynamic climate-driven controls on the deposition of the Kimmeridge Clay Formation in the Cleveland Basin, Yorkshire, UK**

Elizabeth Atar[1], Christian März[2], Andrew Aplin[1], Olaf Dellwig[3], Liam Herringshaw[4], Violaine Lamoureux-Var[5], Melanie J. Leng[6], Bernhard Schnetger[7], Thomas Wagner[8].

[1] Department of Earth Sciences, Durham University, South Road, Durham, DH1 3LE, UK
[2] School of Earth and Environment, University of Leeds, Leeds, LS2 9JT, UK
[3] Leibniz-Institute for Baltic Sea Research, Marine Geology, Seestrasse 15, 18119 Rostock, Germany
[4] School of Environmental Sciences, University of Hull, Hull, HU6 7RX, UK
[5] IFP Energies Nouvelles, Geosciences Division, 1 et 4 Avenue de Bois-Préau, 92852 Rueil-Malmaison Cedex, France
[6] NERC Isotope Geosciences Laboratory, British Geological Survey, Nottingham NG12 5GG, UK and Centre for Environmental Geochemistry, School of Biosciences, University of Nottingam, Sutton Bonington Campus, Leicestershire, UK
[7] ICMB, Oldenburg University, P. O. Box 2503, 26111 Oldenburg, Germany
[8] Lyell Centre, Heriot-Watt University, Edinburgh, EH14 4AS, UK

**Abstract**

The Kimmeridge Clay Formation (KCF) is a laterally extensive, total organic carbon-rich succession deposited throughout Northwest Europe during the Kimmeridgian–Tithonian (Late Jurassic). ~~Here we present a petrographic and geochemical dataset for a 40 metre thick section of a well-preserved drill core recovering thermally immature deposits of the KCF in the Cleveland Basin (Yorkshire, UK), covering an interval of approximately 800 kyr. The new data are discussed in the context of depositional processes, sediment source and supply, transport and dispersal mechanisms, water column redox conditions, and basin restriction.~~

[revised manuscript text omitted]
. ~~In contrast, the HVMIs contain marine organic material and carbonate rich faecal pellets. The occurrence of organo-mineral aggregates and algal OM (20 μm thick and 200 μm long; Fig. 3) demonstrate marine snow and algal mat settling as key mechanisms for the delivery of OM to the seafloor (Macquaker et al., 2010). Filter feeding organisms strip nutrients and fine grained sediment out of the water column and excrete them as faecal pellets (Ittekkot et al., 1992). This biological mediation of the sediment is a key process in the export of OM, sediment, and nutrients from the water column to the sediment. Ittekkot et al. (1992) demonstrated a link between enhanced sediment and nutrient flux to the ocean during wet phases of the monsoonal cycle. They showed also that there was an increase in biotic abiotic pellet production, and thus an~~

enhanced OM and mineral flux to the seafloor, in response to enhanced continental runoff and weathering associated with the monsoonal wet phases.

[revised manuscript text omitted]